# Isotopic evolution of planetary crusts by hypervelocity impacts evidenced by Fe in microtektites

S. M. Chernonozhkin [1✉], C. González de Vega[1], N. Artemieva [2,3], B. Soens[4], J. Belza[1], E. Bolea-Fernandez[1], M. Van Ginneken[5], B. P. Glass[6], L. Folco [7,8], M. J. Genge [9], Ph. Claeys [4], F. Vanhaecke [1] & S. Goderis [4✉]

Fractionation effects related to evaporation and condensation had a major impact on the current elemental and isotopic composition of the Solar System. Although isotopic fractionation of moderately volatile elements has been observed in tektites due to impact heating, the exact nature of the processes taking place during hypervelocity impacts remains poorly understood. By studying Fe in microtektites, here we show that impact events do not simply lead to melting, melt expulsion and evaporation, but involve a convoluted sequence of processes including condensation, variable degrees of mixing between isotopically distinct reservoirs and ablative evaporation during atmospheric re-entry. Hypervelocity impacts can as such not only generate isotopically heavy, but also isotopically light ejecta, with $\delta^{56/54}$Fe spanning over nearly 5‰ and likely even larger variations for more volatile elements. The mechanisms demonstrated here for terrestrial impact ejecta modify our understanding of the effects of impact processing on the isotopic evolution of planetary crusts.

[1] Atomic & Mass Spectrometry – A&MS Research Unit, Department of Chemistry, Ghent University, Campus Sterre, Krijgslaan 281 – S12, BE9000 Ghent, Belgium. [2] Planetary Science Institute, Tucson, AZ 85719, USA. [3] Institute for Dynamics of Geospheres RAS, 117334 Moscow, Russia. [4] Analytical, Environmental, and Geochemistry, Vrije Universiteit Brussel, Pleinlaan 2, BE1050 Brussels, Belgium. [5] Centre for Astrophysics and Planetary Science, School of Physical Sciences, Ingram Building, University of Kent, Canterbury CT2 7NH, UK. [6] Department of Earth Sciences, University of Delaware, Newark, DE 19716, USA. [7] Dipartimento di Scienze della Terra, Università di Pisa, 56126 Pisa, Italy. [8] CISUP, Centro per l'Integrazione della Strumentazione dell'Università di Pisa, 56126 Pisa, Italy. [9] IARC, Department of Earth Science and Engineering, Imperial College London, Exhibition Road, London SW7 2AZ, UK. ✉email: Stepan.Chernonozhkin@Gmail.com; Steven.Goderis@vub.be

Evaporation and condensation are fundamental processes in the chemical evolution of the Solar System. When the protosolar dust cloud evolved into the rocky terrestrial planets, the asteroids and planetesimals accreting to these planets were significantly depleted in moderately volatile elements as the result of these processes[1]. Collision was an important source of heat in the early Solar System potentially also leading to evaporation, in conjunction with radioactive decay and gravitational heating[2]. The isotope ratios of the moderately volatile elements are sensitive to evaporation and condensation. Altered isotopic compositions of these elements are observed for the Moon[3], formed as the result of a giant impact, for the volatile-depleted parent asteroids of the ureilite[4] and angrite[5] meteorites, for asteroid Vesta[6], and potentially also for Mercury with its partially stripped mantle[7].

Terrestrial ejecta, which formed during hypervelocity impacts that were especially prevalent during the early Solar System, are particularly interesting for studying the effects of evaporation that occurred during these events. Such impacts generate ejected silicate melt with broadly upper crustal compositions as a result of extreme heat and pressure. Tektites are distal impact glasses formed by melting of terrestrial rocks upon hypervelocity meteorite impacts[8]. Microtektites are the sub-millimeter analogs of tektites, thought to have solidified from smaller droplets of melt. Both tektites and microtektites are ejected from the source crater, travel ballistically, are quenched in flight, and are deposited hundreds to thousands of kilometers away from the source crater, in so-called strewn fields. A key feature of tektites is their volatile-depletion, observed for water, as well as for volatile and moderately volatile elements[9].

Of the four historically recognized strewn fields, the Australasian tektite and microtektite strewn field is the largest and most recent example. Australasian tektites and microtektites are linked to a hypervelocity impact origin by their geochemical compositions and features characteristic for shock melting: the absence of primary minerals, the presence of lechatelierite and shocked mineral grains in a subset of tektites and microtektites[10], as well as the occurrence of high-pressure mineral phases in some tektites[11]. The recovery of shock metamorphosed rock and unmelted mineral fragments in Australasian microtektite layers in the South China Sea[12] further supports the impact origin. Australasian microtektites have been collected from deep-sea sediment cores from the Ocean Drilling Program (ODP), Deep Sea Drilling Project (DSDP), as well as from piston cores by a variety of oceanographic institutes in over 60 locations in the Indian and western Pacific Oceans and adjacent seas[12,13], and from several sediment accumulation traps in Antarctica[14–17].

Iron is a major rock-forming element in most planetary reservoirs with variable geochemical behavior in an assortment of contexts. Iron isotope ratios have been used as a key tracer of high-temperature Solar System processes, but also as a useful tool to study the geochemical cycle of Fe in low-temperature environments. The Fe isotopic composition of Earth's mantle rocks ($\delta^{56/54}Fe = 0.00 \pm 0.22‰$, 2SD) is close to the Solar System average ($\delta^{56/54}Fe = -0.008 \pm 0.095‰$, $n = 47$ CI chondrites)[18]. The isotopic composition of Fe in the upper continental crust, as estimated from the composition of loess-paleosol sequences[19], is slightly heavier ($\delta^{56/54}Fe = 0.09 \pm 0.03‰$, 2SD). Terrestrial basalts have $\delta^{56/54}Fe$ of $0.08 \pm 0.13‰$ (2SD), while differentiated $SiO_2$ rich rocks display heavier Fe isotopic values in the range of 0.1–0.5‰[18]. Iron is a comparatively refractory element (1338 K[20] 50% condensation temperature under nebular conditions, $10^{-4}$ atm, and 2830 K[21] 1% evaporation temperature from silicate melts to air). Thus, energetic events are required for evaporation to play a significant role in the Fe isotope budgets of planetary bodies. Since early reports on Fe isotopic compositions of the rocky bodies in the inner Solar System, it has been suggested that the observed differences between the Moon, Earth, Mars, and some asteroids may result from variable degrees of evaporation experienced during their evolution[22]. Although since then, it has become clear that the Fe isotopic composition of planetary surfaces is affected by both core-mantle segregation[23,24] and redox-dependent[25] mantle differentiation[26,27], these processes alone may not be sufficient to account for the observed signatures of meteorites[28]. As such, evaporation remains a viable process that could affect the Fe isotope budgets during planetary evolution either at the nebular stage, during the accretion, or subsequently following episodes of hypervelocity impacts or magmatism[28,29]. Fractionation of Fe isotopes in impact plumes was observed earlier in the case of metal grains of CB chondrites[30]. The process of thermal evaporation of a target rock is typically associated with a loss of volatile components, shifting the residual melt towards heavier isotopic compositions.

Isotope fractionation due to partial distillation during impact processes has a direct bearing on our understanding of key processes in the planetary sciences, such as asteroid or planetary collisions, for example during accretion or the Moon-forming impact[3]. Because lighter isotopes of the volatilized element have incrementally higher evaporation rates than their heavier counterparts, the vapor phase is enriched in the lighter isotopes, while the residual fraction is progressively more enriched in the heavier isotopes. As a result of evaporation, the concentration of the moderately volatile elements in the residual phase is reduced and their isotopic composition shifted towards heavier values[1,31,32]. Tektite glass represents a natural laboratory, which might be used to study impact-related Fe isotope fractionation. The isotopes of other moderately volatile elements have previously been shown to fractionate in tektites[4,33–40]. Of these elements, Cu, Zn, and Sn systematically display heavy isotopic signatures (~10‰ variation for $\delta^{65}Cu$, ~3‰ for $\delta^{66}Zn$, and ~2.5‰ for $\delta^{122}Sn$), which correlate with the extent of element depletion as a result of evaporative loss[4,33–36]. The Cd isotopic composition of a single tektite from Laos[37] displayed an isotopic composition 0.76‰ per amu heavier than the average terrestrial composition. At the same time, other volatile elements exhibit no isotope fractionation due to evaporation in macroscopic tektites. As an example, the isotopic variations of Li and K only reflect mixing of distinct reservoirs within an inhomogeneous target[34,38–40].

Due to their sub-millimeter size, microtektites have different surface-to-volume ratios and heating histories as a function of time, as compared to macroscopic tektites. While macroscopic tektites were studied intensely for potential evaporative isotope fractionation, only a single study so far has focused on microtektites, reporting K isotope ratio data for 28 Australasian microtektites[41]. The observation of extreme K isotope fractionation in microtektites, with no significant variations in their macroscopic analogs[34], implies that microtektites may be more sensitive to evaporation in hypervelocity impact processes.

The main aim of this work is to assess and refine the mechanisms of isotope fractionation taking place during hypervelocity impact events and ejecta emplacement by studying the Fe isotopic compositions of Australasian microtektites. To determine the Fe isotopic compositions of these microscopic glasses in situ, we use a laser ablation unit coupled to a multicollector ICP mass spectrometer. The mechanisms of evaporation during impact events currently remain poorly understood. Thus, the composition of the microscopic ejecta particles formed in a single impact event 790 ka ago may improve our understanding of the effects of collisions on the isotopic evolution of planetary crusts, and may ultimately improve the use of Fe as a proxy of planetary processes.

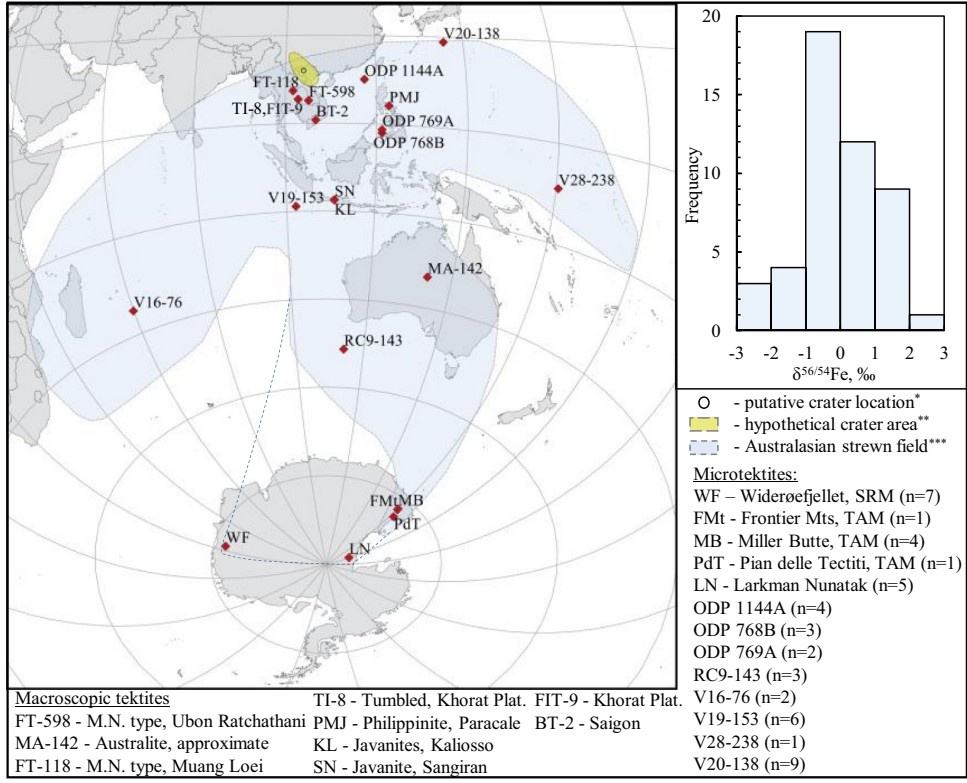

**Fig. 1 Collection sites of the tektites and microtektites studied.** The microtektites are collected from the sea sediments and sedimentary traps in Antarctica. *n* corresponds to the number of microtektites studied from each location. Top right – frequency distribution plot for $\delta^{56/54}$Fe values of the microtektites. *22° N and 104° E is one of the proposed crater locations based on the abundance of unmelted ejecta and microtektites[12]. **A location of the impact crater anywhere within the yellow oval explains the distribution of the microtektites and impact ejecta equally well[12]. ***Australasian strewn field after Folco et al[80]. extended to the Transantarctic Mountains[12]. The blue dashed line suggests an extension of the central lobe to the Widerøefjellet, SRM[16]. M.N. Muong Nong tektite shape type, WF Widerøefjellet, SRM Sør Rondane Mountains, Antarctica, FMt Frontier Mountains, Antarctica, TAM Transantarctic Mountains, Antarctica, MB Miller Butte, Antarctica, PdT Pian delle Tectiti, Antarctica, LN Larkman Nunatak, Antarctica, ODP microtektites from ocean sediments collected during Ocean Drilling Programs.

## Results

**General observations**. Here, we study a collection of 48 microtektites from 14 locations in the Indian and Pacific Oceans and Antarctica, and 13 macroscopic tektites from Thailand, Vietnam, the Philippines, Indonesia and Australia, distributed across the Australasian strewn field (Fig. 1). All microtektites analyzed in this work are glassy spherules with a diameter of 60–600 µm. Petrographic observations suggest that all particles studied are relatively unaltered, with no cracks or signs of alteration (see the supplementary materials). The concentrations of major and trace elements are within the range previously reported for Australasian macroscopic and microtektites (Fig. 2).

**Iron isotopic compositions**. Iron isotope ratios, expressed as δ56/54Fe, range from −2.85 to +2.19‰, with an average uncertainty of 0.1‰ (1SD). A frequency distribution of the Fe isotope ratio variation is presented in Fig. 1. Iron isotopic signatures show a covariation with the average distance from the presumed impact crater to the site location, the most distal Antarctic microtektites containing isotopically heavier Fe (Fig. 3). Note that a crater location at 22° N and 104° E[12] was used here to calculate great circle distances, although other suggested locations[42] would provide nearly equal distances. Smaller microtektites tend to display isotopically lighter Fe isotopic signatures (Fig. 3). Microtektites 137-4 from RC9-143 and 17 from ODP 769A exhibit FeO contents of 9.01 and 8.29 wt%, respectively, nearly twice as high as the average value for normal-type Australasian microtektites[15,43]. These high contents may reflect a

distinct endmember composition within the target lithologies or an immiscible Fe-rich melt known to occur within specific tektites[14,43,44]. These particles are not discussed together with the other microtektites, as they likely represent distinct endmember compositions in the precursor material. For the numerical data, the reader is referred to the appendix.

## Discussion

Evaporative isotope fractionation has long been suggested as the process that governs the isotopic signatures of moderately volatile elements (e.g. Zn, Cu, Cd, Sb) in macroscopic tektites, without additional details on the exact mechanism[33,35]. The results of this work show no significant variation in the Fe isotopic composition of macroscopic tektites ($\delta^{56/54}$Fe between −0.30 and +0.14‰), suggesting that evaporation did not play a significant role in the evolution of their Fe isotopic signatures. The $\delta^{56/54}$Fe values and FeO concentrations in microtektites, found in ODP sediment cores drilled in the South China and the Sulu Seas indicate only marginally significant Fe isotope ratio variability. The $\delta^{56/54}$Fe values cluster near average continental crustal values from −0.46 to +0.29‰. The measured data for the microtektites from these sites are also not consistent with the trends expected for evaporative fractionation. Relative to the other microtektites analyzed in this work, the microtektites from the South China and the Sulu Seas were deposited closer to the hypothetical impact crater location (1300–2350 km[12]). Their trace element signatures indicate minimal, if any, evaporation following their residence in the impact plume and ballistic transfer (Fig. 2). Farther away

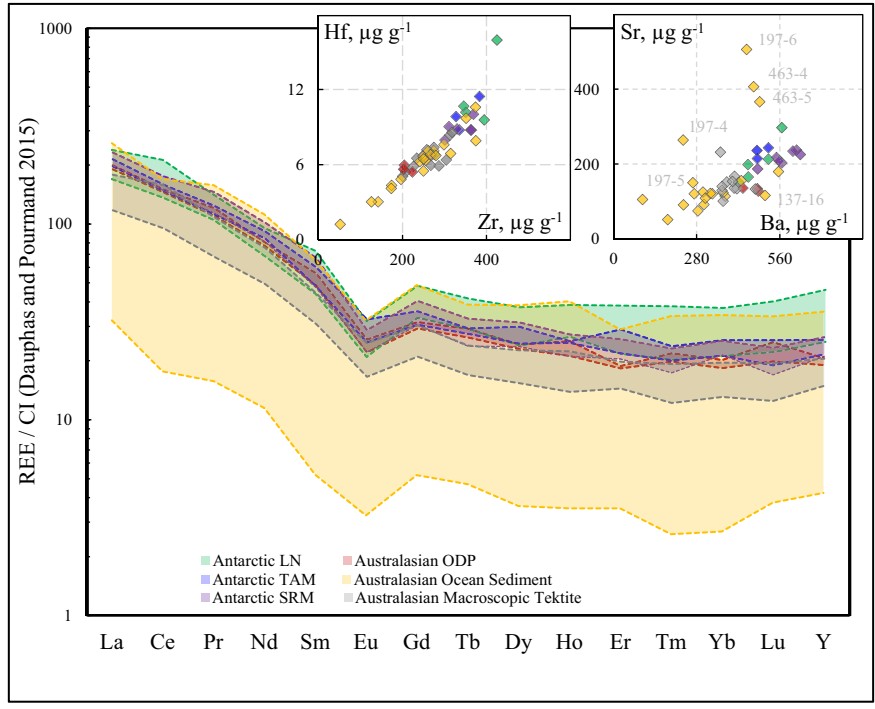

**Fig. 2 REE signatures of the tektites and microtektites studied.** REE concentrations are normalized to CI-chondrite values[84]. LN Larkman Nunatak, Antarctica, TAM Transantarctic Mountains, Antarctica, SRM Sør Rondane Mountains, Antarctica, Widerøefjellet, ODP microtektites from ocean sediments collected during Ocean Drilling Programs 768B, 769 A and 1144 A, Ocean Sediments – microtektites collected from piston cores during oceanographic studies.

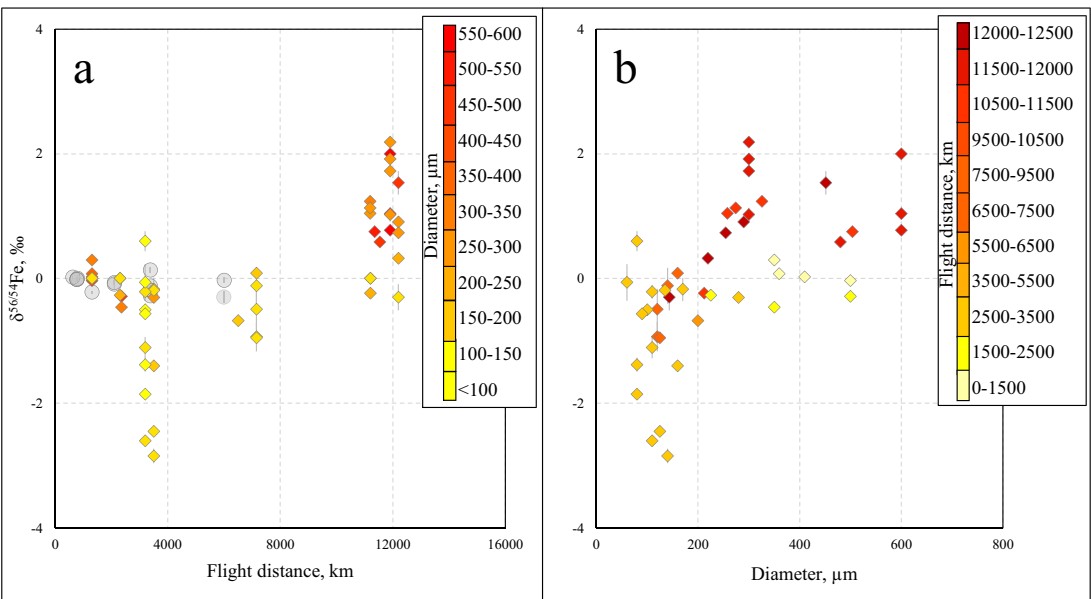

**Fig. 3 Relations between Fe isotopic signatures, distance, and size. a** - Fe isotopic signatures of the Australasian microtektites versus their distance to the hypothetical crater location. **b** – Fe isotopic signatures of the microtektites versus their diameter. Macroscopic tektites are shown using gray circles.

from the putative crater are the Antarctic microtektites from Widerøefjellet in the Sør Rondane Mountains (SRM), which traveled up to 12000 km. These microtektites display the highest $\delta^{56/54}Fe$ values ranging between +0.77 and +2.19‰ and are associated with lower FeO contents (Fig. 4). The Fe isotope ratio data of the SRM microtektites are consistent with a system that follows a Rayleigh relationship with a starting composition characterized by $\delta^{56/54}Fe \approx 0$ and FeO concentration of 6.9 wt%

(consistent with average crust), $f$ equal to 0.71 – 0.58 and a fractionation factor α between 0.995 and 0.997. However, the Fe isotopic signatures of Australasian microtektites do not simply depend on the distance from the crater (Fig. 3). For example, the Australasian microtektites from RC9-143, one of the most remote locations from the putative crater evaluated in this work, do not display heavy Fe isotopic signatures, which might be assigned to evaporation. The isotopic variability at a particular site (and

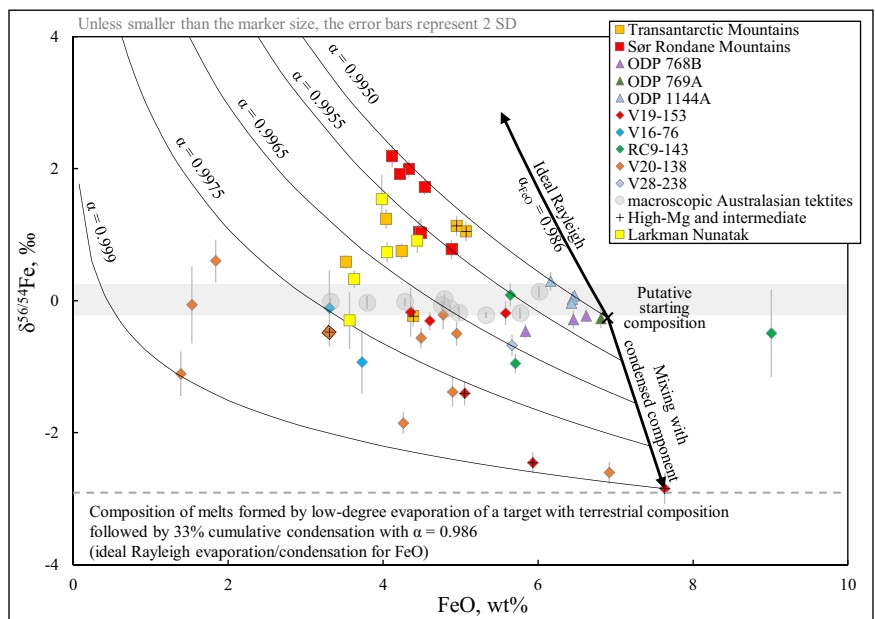

**Fig. 4 Diagram of δ⁵⁶/⁵⁴Fe versus FeO.** The shaded horizontal area indicates the Fe isotopic composition in macroscopic Australasian tektites. The average composition of the upper continental crust (FeO = 6.9 wt%, δ⁵⁶/⁵⁴Fe = −0.263‰) suggested to be the starting composition for the Australasian strewn field. A range of Rayleigh evaporation trends predicted using different fractionation factors α are shown. The Rayleigh fractionation trend under ideal conditions of evaporation to vacuum predicted by the Hertz–Knudsen equation is shown as a thick line. The downward shift of certain microtektites from the upper crust δ⁵⁶/⁵⁴Fe–FeO composition is best explained by a mixing line between the pristine melt and condensed components.

distance) from the crater may be caused by the analysis of an unrepresentative number of particles, preventing the derivation of evaporation trends. On the other hand, the restriction of heavy Fe isotopic signatures to the more distally deposited Australasian microtektites provides clear evidence that evaporative Fe isotope fractionation does take place during hypervelocity impact events, while the magnitude of this fractionation is among the largest observed for Fe in silicate materials from natural environments. The microscopic size thus plays a critical role in the isotope fractionation mechanism of microtektites, because they experienced different peak temperatures and temperature profiles over time, compared to their macroscopic analogs due to different surface-to-volume ratios.

The distinct Fe isotopic signatures of microtektites between more proximal (light) and distal (heavy) collection sites, with Antarctic microtektites having experienced the largest evaporation, indicates Fe isotope fractionation in the hotter and cooler areas of the impact plume. It has previously been shown that impact glass particles that form during the earliest stages of the impact cratering process, in the hottest and fastest part of the impact plume closer to the growing transient cavity, undergo more evaporation and homogenization, are smaller, and contain less vesicles and mineral inclusions[10]. Particles originating from hotter zones in the impact plume experienced larger degrees of evaporation and have at the same time been accelerated to higher velocities, leading to their deposition farther from the impact crater. This evaporation mechanism, previously demonstrated for a collection of Australasian microtektites and for microtektites from the Larkman Nunatak[17], in which the abundances of the volatile alkali elements decrease progressively with the flight distance[45], appears to be similarly applicable to the Fe isotope fractionation observed in this work.

While the extreme conditions of hypervelocity impact events are likely the primary factor controlling the evaporation and isotope fractionation in microtektites, ablative evaporation during flight and interaction with the atmosphere may also take place. Microtektites have cooled quickly after pressure release and

plume expansion, so by the time they reach space, the evaporation ceases. Different from the much faster cosmic spherules[46], gas drag cannot contribute to heating of the microtektites over most of their ballistic trajectories, as they escape the atmosphere. However, our model based on standard equations for a meteoroid entry with drag and ablation (see Materials and Methods) indicates that the most remote Australasian microtektites, which reach Antarctica, must have experienced velocities and trajectory angles that could lead to partial vaporization and mass loss during atmospheric re-entry. As shown in Fig. 5, the temperature and mass loss due to vaporization depend on the velocity and size of the particle. Particles with a diameter of 150–300 μm that reach Antarctica may lose 40–80% of their initial size during atmospheric re-entry (Fig. 5, blue curves), while particles smaller than 150 μm experience nearly no evaporation. At the same time, proximal Australasian microtektites display lower velocities when they re-enter the atmosphere, are decelerated more quickly due to their small masses, and are therefore not reheated to evaporation temperatures. As such, they are not vaporized by friction in the atmosphere and preserve their initial volatile contents and Fe isotopic compositions, acquired during the impact event itself. Compared to most cosmic spherules, the mass loss experienced by the most distal Australasian microtektites during atmospheric re-entry appears to be significantly lower, as reflected by the absence of Ce anomalies in their REE patterns (Fig. 2). Cerium anomalies are occasionally observed in cosmic spherules as a result of oxidative conditions during evaporation[47].

The light Fe isotopic signatures observed for a fraction of the microtektites however require a different mechanism. Whilst most Antarctic microtektites are fractionated towards heavier Fe isotopic signatures, a small number of microtektites from the Transantarctic Mountains (TAM) and Larkman Nunatak (LN) do not exhibit positively fractionated values. This may suggest that the SRM microtektites underwent slightly different processes during the impact event compared to the Antarctic microtektites from the TAM and LN sites. Alternatively, TAM and LN microtektites could contain a higher proportion of an isotopically

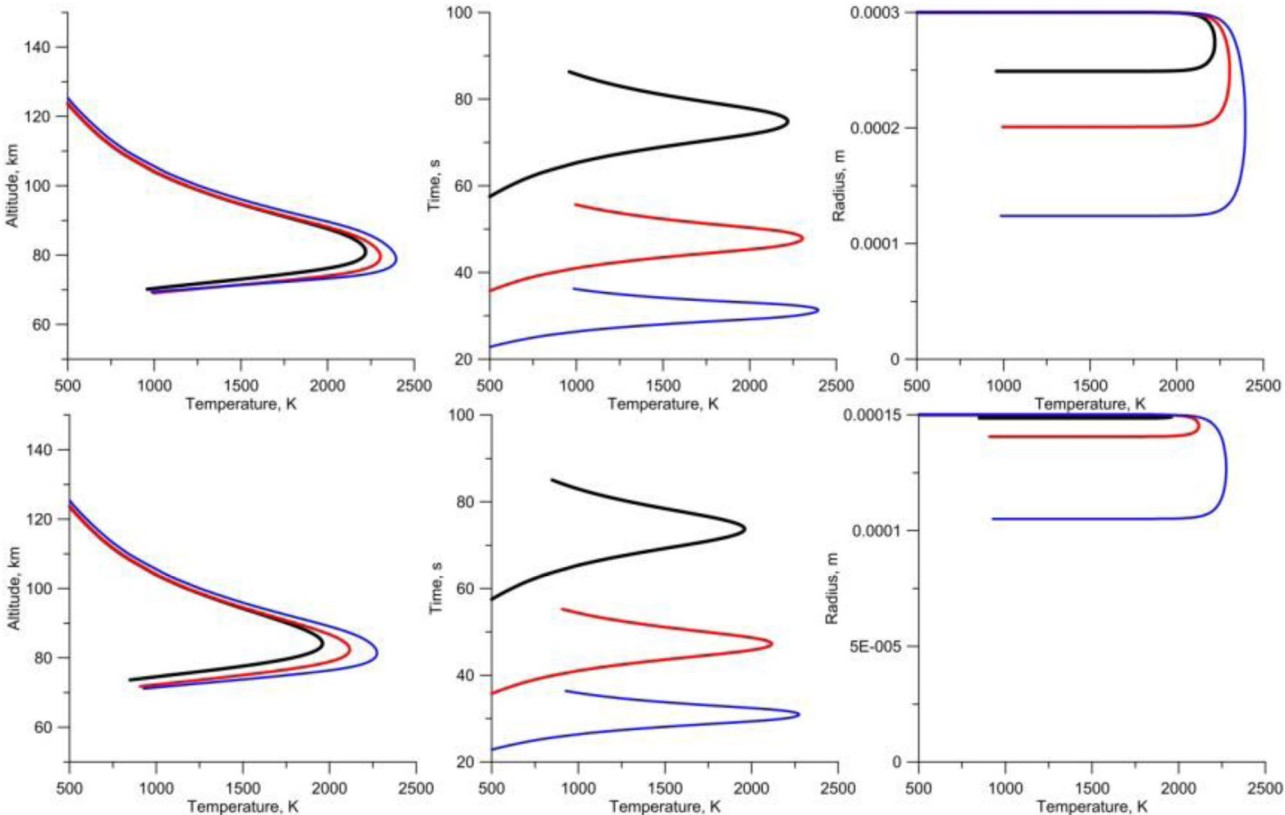

**Fig. 5 Ablation of Antarctic microtektites during re-entry.** Particles entering the atmosphere with different velocities and angles can heat and ablate. Upper plots are for a particle radius of 300 μm, the lower ones for a particle radius of 150 μm. Black: velocity of 7.4 km s$^{-1}$, re-entry angle of 15°; red: 7.4 km s$^{-1}$, angle of 27°; blue: 8 km s$^{-1}$, angle of 42°. All combinations of angle/velocity result in deposition at ~11,000 km from the source crater.

light Fe component, a product of condensation, relative to those from the SRM. However, the major and trace element geochemistry of these groups does not record such processes or otherwise these processes led to similar concentrations. Similarly, the variation of nearly ~1‰ in δ$^{56/54}$Fe observed for ODP microtektites must be related to a heterogeneously distributed isotopically light phase during the turbulent impact plume stage. Turbulence is an important property inherent to the structure of impact plumes, and thus perfect geochemical trends with distance are not to be expected in general.

Surprisingly light Fe isotopic signatures down to a δ$^{56/54}$Fe value of −2.85‰, observed for microtektite 137-8 from the V19-153 location in the Eastern Indian Ocean, cannot be explained based on a continuously evaporating melted reservoir. These light signatures are also unlikely to represent initial isotopic heterogeneity within the target material[48]. Most terrestrial (igneous) rocks are characterized by δ$^{56/54}$Fe values close to zero[18]. Certain sulfides[49] and carbonates[50,51] display light Fe isotopic compositions, but a sulfide- or carbonate-rich target cannot explain the Fe signatures of microtektites based on the bulk geochemistry and Fe isotopic signatures of macroscopic tektites. The Australasian microtektites studied here are highly pristine, and no weathering features, which could alter the Fe isotopic signatures, are observed. The major and trace element concentrations measured earlier and in this work[12,16,17,45] confirm that limited to no alteration affected the microtektites (Fig. 2). For a detailed discussion on the potential mechanisms that could lead to light Fe isotopic signatures other than condensation, the reader is referred to the appendix.

On the other hand, condensates with light isotopic compositions are typical for condensation under variable kinetic regimes. For example, condensation of isotopically heavy droplets can progressively shift the gas phase towards light isotopic signatures. Over time, the instant condensate can become isotopically lighter compared to the initial isotopic composition[52]. Light isotopic signatures may also result from low degrees of thermal volatilization of the target followed by rapid nearly complete recondensation of the vapor. Various mechanisms following condensation of oversaturated gas also remain possible, as this process is highly sensitive to pressure and cooling rate[31].

Thus, the extremely light Fe isotopic signatures of microtektites down to δ$^{56/54}$Fe of −2.85‰ most likely result from interaction of the ejected melt of the target with isotopically light recondensed melts. Potentially, the condensation of a hot oversaturated gas phase starts during the expansion of the impact plume when the pressure and temperature are rapidly reduced. Interaction with condensed melts for Australasian microtektites has previously been advocated to explain the occurrence of Fe,Ni,Cr-Ti-rich spinel grains on the external surface of some microtektites[43], and the full range of K isotopic compositions measured for microtektites.[41] Variations in the K isotopic composition up to percent levels have previously been measured among a set of 13 Australasian microtektites[41] (down to δ$^{41}$K = −10.6‰), both in positive and negative direction, which is in full agreement with the Fe isotope fractionation observed here. The observation that microtektites closer to the source location contain slightly less K relative to macroscopic tektites is consistent with the incorporation of a condensed component in Australasian microtektites[39,45]. In the case of macroscopic tektites, the effect of condensation is likely insignificant as a result of their low surface-to-volume ratios[53], and light isotopic signatures of Fe and K have not been observed to date.

These observations provide strong evidence that condensation during the impact event played an important role in the

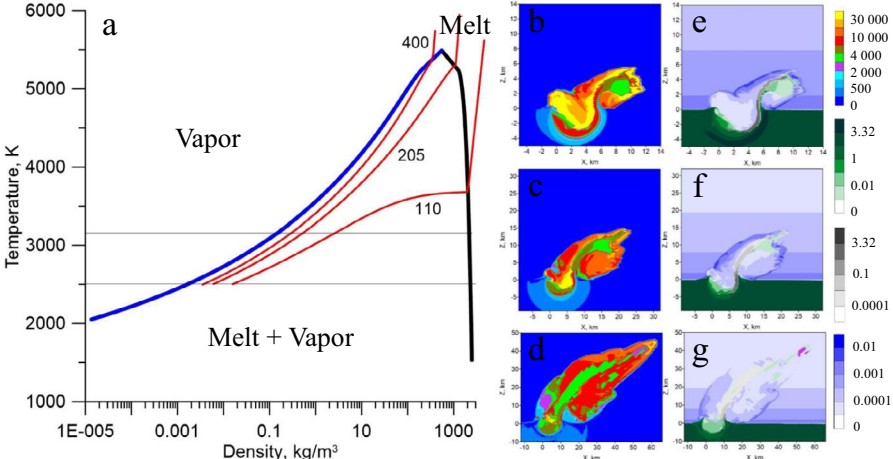

**Fig. 6 Temperature and density in the Australasian impact plume. a** - black and blue lines show the phase boundaries of quartz as calculated by the ANEOS package. The black branch represents saturated liquid, while the blue branch denotes saturated vapor. Red lines show release adiabats after shock compression to 110, 205, and 400 GPa. Thin horizontal lines correspond to the atmospheric pressure at sea level ($10^{-4}$ GPa) and at an altitude of 30 km ($10^{-6}$ GPa). The ejected materials with shock pressures below ~300 GPa, start to boil when the pressure is released. When the shock compressions are above ~300 GPa, the ejected material condenses during pressure release. The fraction of vapor remaining after the release is 0.1, 0.28, and 0.45, respectively. **b**–**d** –Temperature cross-sections of the plume through the plane of symmetry XZ at time frames of 0.5, 1.5, and 4.6 s following the impact at 45°. **e**–**g** – density cross sections of the impact plume at the same timeframes. Target materials are shown in green, projectile in gray and the atmosphere in blue.

formation of at least a fraction of the ejecta produced. Moreover, microtektites indicate overprinting of heavy Fe isotopic signatures by light vapor condensate.

The relation between the Fe isotopic signatures and FeO contents of the Australasian microtektites, including those from Antarctica, implies more convoluted processes driving their Fe isotope fractionation beyond single Rayleigh-style evaporation[1,31] or mixing with an isotopically light component in the impact plume. Evaporation is not restricted to Antarctic microtektites only, as one microtektite from ODP 769A, another particle from ODP 1144A, and a third one from V20-138 also display slightly positive $\delta^{56/54}$Fe values. However, most microtektites from the Indian and Pacific Oceans demonstrate signatures approximating zero or display negative $\delta^{56/54}$Fe values, suggesting more complex fractionation histories. The microtektites from the V20-138 and V19-153 locations in the Pacific and Indian Ocean, respectively, both show weakly negatively correlated $\delta^{56/54}$Fe and FeO contents. The Fe isotopic compositions for each of these locations can loosely be explained by Rayleigh distillation expected for thermal evaporation. However, compared to the putative target composition, these trends suggest that the microtektites from V20-138 and V19-153 formed by evaporation from the material that initially had an extremely light Fe isotopic signature ($\delta^{56/54}$Fe between −2.61 and −2.85‰) and an elevated FeO content between 7.19 and 7.32 wt%. Such initial compositions are in agreement with the lightest isotopic signatures preserved among the microtektites measured here, as a result of condensation. If only evaporation was responsible for these signatures, this would imply fractionation factors (α) equal to 0.998–0.9935 and up to 80% FeO loss due to evaporation. In addition, these microtektites indicate stronger scatter along the Rayleigh curve. Such signatures are thus best explained by a combination of (i) isotope fractionation of the melt towards heavier values due to evaporation and (ii) mixing with variable amounts of a re-condensed, isotopically light component.

However, the mixed Fe isotopic signatures cannot be explained by assuming that all microtektites form within the plume as fractional condensates that subsequently underwent partial evaporation during atmospheric re-entry in the case of sufficiently

high velocities. Based on the presence of abundant vesicles and the occurrence of mineral inclusions that display some indications of melting in particular specimens[10], the formation of all microtektites solely by condensation is not possible.

Although evaporation during ballistic transport and interaction with the atmosphere cannot be excluded in the case of the most distally deposited (Antarctic) microtektites, evaporation must also take place earlier on in the impact plume, as suggested for lunar glass beads that form in the absence of an atmosphere[54]. In this case, the evaporation and condensation regimes must coexist in the rapidly expanding impact plume. Within the first few seconds after the impact, molten, shocked and partially vaporized materials from the upper target layers are ejected. The resulting impact plume is a turbulent mixture of vapor, melt, and solid fragments with wide compositional and thermal ranges. The temperature of the ejected materials is mainly defined by the melt-vapor transition of quartz and does not exceed ~4000 K. As the plume expands, the temperature and pressure of the ejected material start to drop quickly along the pressure release adiabat on the phase diagram (Fig. 6a). Depending on the initial shock level of the ejected materials, this material may enter the two-phase area as vapor or liquid. As a result, materials shocked above ~300 GPa start to condense, whereas materials shocked below 300 GPa commence to boil, resulting in different isotopic compositions. In addition to distinct pressure release patterns, the ejected materials display dissimilar cooling rates in different parts of the plume (Fig. 6b). The part of the impact plume expanding downrange initially has the highest temperature and velocity (>10 km s$^{-1}$) and cools down faster than the central part of the plume, which is characterized by lower velocities and a higher density. In this context, the cooling rates are known to be an important factor affecting the final isotopic composition of the condensed material[31].

As an alternative scenario, the isotope fractionation mechanisms may be overprinted in the impact plume when the microtektites initially form as a partially vaporized expulsed melt. These droplets may then serve as nucleation seeds on which an isotopically light component starts to re-condense as the temperature and pressure drop, allowing for condensation (Fig. 6a). Such

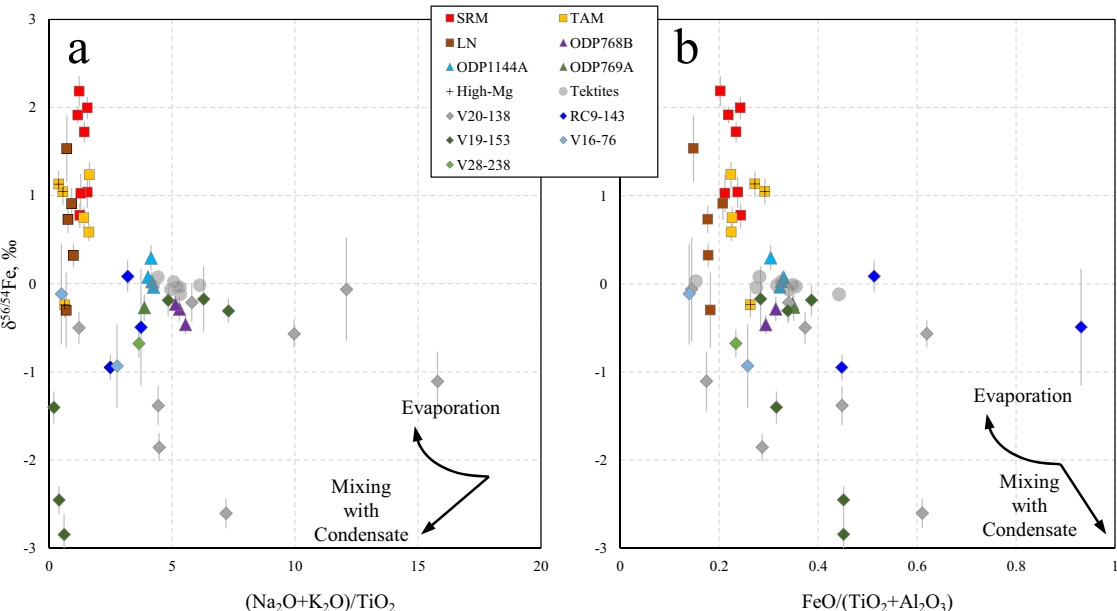

**Fig. 7 Fe isotopic signatures vs bulk compositions. a** – Plot of $\delta^{56/54}$Fe versus the alkali elements normalized by the refractory element Ti to illustrate the effect of evaporation and condensation on volatile elements in microtektites. The Australasian microtektites that experienced the hottest conditions in the impact plume display the lowest concentrations in the alkali elements, together with the highest magnitudes of Fe isotope fractionation, both in positive and negative direction. The Antarctic microtektites experienced additional evaporation during atmospheric re-entry. **b** – Diagram of $\delta^{56/54}$Fe versus FeO normalized to major refractory elements, used to demonstrate that the absolute FeO concentrations are affected by volatile loss of Fe rather than the volatilization of other, more volatile major elements. SRM Sør Rondane Mountains, Antarctica, TAM Transantarctic Mountains, Antarctica, LN Larkman Nunatak, Antarctica.

mechanism of heterogeneous nucleation on the surface of pre-existing particles has previously been demonstrated to occur in volcanic eruption plumes[55]. Similarly, a thin layer of material covering the inner part has been observed for lunar glasses, suggesting heterogeneous condensation in volcanic[56] and impact[57] plumes. As no isotopically light microscopic glass[54,58] was found in the vicinity of the crater, homogeneous condensation may not have been a prevailing mechanism in the plume. Because the impact event produces relatively little material[58] with extremely high compression >300 GPa (Fig. 6) that allows for condensation, the isotopically light reservoir potentially existed only in the form of condensate at the surface of nucleation seeds, and was later mixed and diluted in these seeds. Large scatter around the Rayleigh evaporation trend for microtektites from the V20-138, V19-153, and LN locations might be explained by individual microtektites incorporating variable proportions of condensate diluted in the evaporated melt. This admixture might have taken place when the ballistic paths of these microtektites passed through turbulent and heterogeneous parts of the condensing impact plume. The Antarctic microtektites with the heaviest Fe isotopic signatures appear to have experienced the least interaction with this isotopically light Fe component and/or underwent an additional episode of evaporation during atmospheric re-entry. Among the Antarctic collections, the LN microtektites display the largest within-site variability in $\delta^{56/54}$Fe and the lowest values, suggesting considerable overprinting of the isotope fractionation mechanisms.

The Fe isotopic signatures of microtektites suggest Rayleigh evaporation under non-ideal conditions. Laboratory vacuum evaporation of FeO has previously constrained the ideal fractionation factor $\alpha$, while evaporation of mixtures of oxides resulted in slightly lower fractionation magnitudes than expected for the Rayleigh process[32]. All fractionation factors $\alpha$ fitted to the subsets of the microtektite data differ significantly from that

predicted by ideal distillation to vacuum for both Fe (0.982) and FeO (0.986) species, suggesting a more complex mechanism. This implies non-equal evaporation coefficients $\gamma$ for the Fe isotopes in microtektites, in agreement with a diffusion-limited evaporation regime, as proposed for Cu and Zn in macroscopic tektites[35,36,59]. Alternatively, a model of evaporation from a supercritical fluid during rapid cooling may be invoked, the so-called "bubble-stripping" evaporation mode, which is suppressed at high pressures[34,60]. In the past, local partial re-condensation in the thin layer above the surface of the microtektite during evaporation was used to explain non-ideal Rayleigh distillation[32]. However, due to their supersonic velocity, microtektites are expected to lose such vaporized shell before it re-condenses. The mean free path of an evaporated species in the vapor is likely to be large compared to the particle size, making it unlikely for these species to re-condense. Finally, the deviations of the fractionation factors found to best fit the microtektite data from predictions of ideal evaporation to vacuum may be explained in terms of the Hertz–Knudsen theory by Fe partial pressures that are 64 to 93% of the saturation pressure during evaporation[31]. As ideal Rayleigh fractionation factors $\alpha$ are rarely achieved in natural systems[33–35], the range of fractionation factors ($\alpha = 0.995$–$0.999$) fitted for Fe isotope fractionation in microtektites may be used as a first order estimate to interpret Fe isotope fractionation in terrestrial hypervelocity impact events.

If the envisioned processes affected Fe, these should also have influenced more volatile elements. Refractory elements[32] are often used to normalize the content of the element fractionated in the evaporation process. In the case of the microtektites, such normalization does not simplify the observed patterns, confirming a more convoluted process (Fig. 7). In contrast to the relations between $\delta^{56/54}$Fe and FeO, which can be explained by Rayleigh evaporation trends and mixing of heavy and light re-condensed melt components, the Australasian microtektites that have the

lowest concentrations of volatile elements are characterized by both the lightest and the heaviest Fe isotopic compositions (Fig. 7). Here, the ratio of $Na_2O + K_2O$ normalized to refractory $TiO_2$ is taken as a measure of volatile retention. While progressive evaporation results in a decreased content of volatiles and an enrichment of the heavier Fe isotopes in the melt, the condensed component with isotopically light Fe displays significantly depleted concentrations of Na and K. This is because at the temperatures typical for partial condensation of Fe, alkali elements largely remain in the gaseous phase[61]. Both the most evaporated and re-condensed melts experienced the hottest conditions in the impact plume and are depleted in alkali elements. The microtektites with an Fe isotopic composition close to that of the target rock either did not undergo intense heating, or their initial isotope fractionation by evaporation was balanced out by mixing with condensate enriched in the lighter Fe isotopes. Overprinting of condensation and evaporation results in microtektites that are characterized by Fe isotopic signatures near the starting value but with strong depletions in the alkali elements. In general, volatile elements are expected to display significantly larger ranges in isotope ratios compared to Fe[41]. Isotopic signatures of volatile elements in microtektites have a high potential to unravel the volatilities of elements in realistic impact conditions.

The observations made here have important implications for specific Solar System processes. Evaporation and condensation in the early Solar Nebula are known to play a significant role in the Fe isotope budgets of chondrules, which experienced an episode of evaporation and re-condensation and have a $\delta^{56/54}Fe$ – FeO pattern resembling that of microtektites (Fig. 4)[62,63]. The effects of impact evaporation on Fe are more scarce, e.g. condensation from an impact plume thought to have led to zoned metal in CB chondrites[30]. Although a highly oxidizing and dense atmosphere sets Earth apart from other Solar System bodies, the observation of condensed melts is not unique to terrestrial impact-related materials. Gas-associated spheroidal precipitates (GASP), and their related counterpart, high-Al, Si-poor precipitates (HASP) formed within impact plumes on the Moon as condensates and partial evaporates, respectively[54], and are found as glassy microscopic spheroids in lunar regolith breccias. Evaporate and condensate glasses with structures and compositions similar to GASP and HASP have also been observed in terrestrial zhamanshinite impact glass[64]. Similar signatures of condensation should be searched for in Martian regolith breccia[65]. Although no Fe isotope fractionation due to condensation was observed in bulk lunar breccia[61], GASP and HASP phases are likely to be characterized by strongly fractionated isotope ratios for a range of elements based on their geochemical compositions[66]. Similarly to microtektites, light isotopic signatures for Fe, Cu, Zn, and Cd in lunar orange glass beads[67,68] imply condensation in volcanic plumes during pyroclastic eruptions on the Moon.

Although it remains unclear under which conditions hypervelocity impacts produce isotopically light ejecta, the results of this work demonstrate that impact materials with Fe isotopic signatures both lighter and heavier than the target can be formed. The suggested formation mechanisms of impact ejecta by evaporation, condensation, and mixing of the isotopically distinct reservoirs stand in stark contrast to most current models, where the evaporated reservoir with a light isotopic signature is assumed to be irreversibly lost[4]. Although the average Fe isotopic composition of the microtektites approximates that of the Bulk Silicate Earth, with a near-Gaussian frequency distribution (Fig. 1), the possible observation of a spatial disconnection between the isotopically heavy evaporated and light condensed components (Fig. 3) may have planetary-scale implications. Whether condensation in the impact plumes affected the bulk compositions of

bombarded bodies, or it remains a local feature of the microscopic spherules requires a deeper understanding of the mechanism of isotope fractionation in impact plumes and of the role of an atmosphere. Therefore, the multi-stage convolute process by which tiny distal ejecta particles evolve may be a fundamental control on the isotopic evolution of planetary surfaces by hypervelocity impacts[4–6].

## Methods

**Materials**. The SRM#1-7 Antarctic microtektites from Sør Rondane Mountains, Widerøefjellet, were collected by SG and characterized by BS[16]. A collection of microtektites from ODP 769 (particles 17, 15) and ODP 1144 A cores (particles 35-38), as well as the Antarctic microtektites from TAM (particles 4.6, 20.3, 20.14, 23.01, 20.17, 7.04) were provided by LF[14,15,45]. The Antarctic microtektites from LN are from the collection of MG[17]. Sample collections in Antarctica were organized in compliance with the Antarctic treaty. The microtektites collected from the sea sediments V19-153 (137-7 to 137-12), V16-76 (137-5 and 137-6), RC9-143 (137-1, 137-2 and 137-4), V20-138 (197-1 to 197-6 and 463-3 to 463-6) and V28-238 (188-3), as well as the microtektites from the ODP 768B core (ODP#1, ODP#3, ODP#4), are part of the collection of BG[69]. Several of these microtektites were characterized in terms of their petrology and geochemistry in previous studies[14–17,45,69]. The macroscopic tektites are collected within the Australasian strewn field in Philippines (PMJ-20 - Paracale), Indonesia (JAV-32 - Sangiran, JAV1.1, JAV1.5, JAV-1.8, JAV0.9 - Kaliosso), Thailand (FT-598 - Ubon Ratchathani, FT-118 - Muang Loei, T18A,B - Khorat Plateau), Vietnam (Saigon BT-2), and Australia (MA-142)[44]. The macroscopic Javanites JAV1.1, JAV1.5 JAV1.8, JAV0.9, Saigon BT-2 and a single Philippinite are from the VUB collection[44]. The macroscopic Philippinite PMJ-20 and MA-142 Australite flanged button were purchased from a commercial vendor. The Muong Nong tektites FT-598 and FT-118, tumbled Indochinites TI8A and TI8B and the Javanite JAV-32 are from the collection of Darryl Futrell. The latitudes and longitudes for the collection sites are presented in the ESI.

**Fe isotopic analysis**. LA-MC-ICP-MS has proven valuable to provide high-precision Fe isotope ratio data in the geosciences, when either spatially resolved information is needed, or when the size of the target restricts its analysis by a conventional solution approach[46,70,71]. In situ high-precision Fe isotope ratios in geosciences are often measured on SIMS[72]. Although SIMS has somewhat superior spatial resolution and its isotope ratio measurement precision is nearly similar to that of LA-MC-ICP-MS, the latter is less prone to matrix and topography effects and has more flexibility for instrumental mass discrimination correction. In this work, Fe isotope ratios of the Australasian macroscopic tektites and microtektites were measured at A&MS-UGent by using a setup consisting of a Teledyne CETAC Technologies Analyte G2 laser ablation system, based on a 193 nm ArF*excimer laser with a pulse duration <5 ns and equipped with a HelexII double-volume ablation cell, coupled to a Thermo Scientific Neptune MC-ICP-MS unit, equipped with a large volume jet interface pump and high-efficiency jet interface. The instrumental mass discrimination was corrected for by a combination of internal standardization using a co-nebulized Ni standard solution and external correction using USGS BHVO-2G or BCR-2G glass reference materials as external standards measured in a sample-standard bracketing sequence. MPI-DING glass reference materials ML3B-G, ATHO-G, GOR128-G, and StHS6/80-G were used for daily validation of isotope ratio accuracy and precision. The analysis was performed by ablating along a line with a laser spot size of 10–30 μm and a laser repetition rate of 10–30 Hz. Validation of the protocol was performed using MPI-DING and USGS glass reference materials and cosmic spherules, which have highly variable matrices and Fe contents. The results of LA-MC-ICP-MS were in full agreement with the Fe isotopic signatures found in literature and with the results obtained following acid digestion, chromatographic Fe isolation and MC-ICP-MS measurement, independent of matrix compositions and the magnitude of the Fe signatures. LA-MC-ICP-MS $\delta^{56/54}Fe$ values of the glass GRMs and cosmic spherules, when plotted versus their $\delta^{56/54}Fe$ values measured after acid digestion, fall on a 1:1 line (0.9561 ± 0.0080) with MSWD = 1.7[46]. For the in-length description of the protocol and the validation of the method, the reader is referred to previous work[46].

**Element concentration measurements**. Concentrations of the major elements in microtektites were measured at A&MS-UGent by using a setup consisting of the Analyte G2 laser ablation system coupled to an Agilent Technologies 8800 tandem ICP-MS instrument. The mass spectrometer was operated in MS/MS mode and the collision/reaction cell was pressurized with $H_2$ to minimize spectral interferences otherwise hampering accurate measurements of elements in the low mass range. Quantification of the major elements was performed relying on external calibration against MPI-DING and USGS glass reference materials combined with normalization of cumulative element oxide level to 100%[73]. The heavier trace element compositions were measured in a separate LA-ICP-MS run with the collision/reaction cell operated in vented mode (as spectral interferences are known to be less severe above the mass-to-charge ratio of 81) and by focusing the laser beam to a larger spot with a diameter of 80 μm in order to achieve higher signal intensities.

**Table 1 Constants used for numeric modeling.**

|  | Microtektites (fused silica) |
|---|---|
| A | 12.856 |
| B | 24 824 |
| $\mu$ | 60 |
| $H_v$, kJ·g$^{-1}$ | 6.4 |
| Density, kg·m$^{-3}$ | 2300–2800 |

These parameters are used for numeric modeling of the atmospheric re-entry of the microtektites.

The trace elements were quantified relying on external calibration against MPI-DING and USGS glass reference materials. One of the major elements (Cr or Mn), quantified in a previous run, was used as internal standard. Such approach allows for a solid sample analysis without preliminary knowledge of the concentration of an internal standard.

**Calculation of isotope fractionation due to evaporation**. In the process of evaporation, the isotopic compositions of the remaining melt and instant gaseous phase are calculated using the Rayleigh fractionation equation: $\delta^{56/54}Fe_{melt} = (\delta^{56/54}Fe_0 + 1000) \, f^{(\alpha-1)} - 1000$ and $\delta^{56/54}Fe_{vapor} = (\delta^{56/54}Fe_0 + 1000) \, \alpha f^{(\alpha-1)} - 1000$, where the delta notation reflects the relative difference between the $^{56}Fe/^{54}Fe$ isotope ratios of the melt, vapor and the initial phase from that of an internationally accepted IRMM014 standard in per mil: $\delta^{56/54}Fe = ((^{56}Fe/^{54}Fe)/(^{56}Fe/^{54}Fe)_{IRMM014} - 1) \cdot 1000$. $f$ is the fraction of the $^{54}Fe$ evaporated and $\alpha$ is a fractionation factor that scales to the square root of the masses[32]. The fractionation factor $\alpha$ between the residue and the gas in the case of evaporation to vacuum is defined by the Hertz–Knudsen equation $\alpha = (\gamma_{56}/\gamma_{54})(m_{56}/m_{54})^{0.5}$, where $m$ represents the masses of the Fe isotopes and $\gamma$ the corresponding evaporation coefficients. At first estimation, the ratio of the evaporation coefficients for different isotopes of the same element approximates unity. The delta notations for Cu, Zn, and Sn discussed in the text are calculated in a similar way as for Fe using the $^{63}Cu$, $^{64}Zn$, and $^{118}Sn$ reference nuclides.

**Numeric model of the impact event**. The impact event is modeled with 25 m resolution ($50 \times 50 \times 50$ m cell), the equation of state is based on pure quartz[74] and includes phase transitions. Radiative energy transfer and kinetics of vaporization/condensation are not included. Shock physics code SOVA[75] was used to model the early stage of impact cratering and ejection of potential tektites and microtektites. The code is coupled with the ANEOS[76] package for quartz target material[74] and a dunitic projectile. A projectile 2 km in diameter with an impact velocity of 22 km s$^{-1}$ at 45º to the horizon and porous material at the upper 100 m of the target (sand) were used to model the Australasian impact, which results in a diameter of the transient cavity of about 18–22 km. The initial stage modeled in this work can easily be scaled to any crater size. Moreover, ejection velocities and shock pressures do not depend on the projectile size. Tracer particles are used to track the ejection of tektites.

**Numeric model of the re-entry of microtektites**. A standard set of differential equations was used to model re-entry of high-velocity particles into the atmosphere[77,78]. The drag equation was modified for a molecular flow regime:

$$m\frac{dV}{dt} = -\rho_a S v^2 \tag{1}$$

where m is particle mass, v – velocity, S – particle cross-section, and $\rho_a$ is density of the atmosphere. The incoming energy $E_{in} = 0.5\rho_a S V^3$ is balanced by radiation and vaporization. Assuming that particles are small enough to be isothermal with temperature T and emissivity $\varepsilon$, the radiative energy is $E_{rad} = 4\, S\sigma\varepsilon T^4$, where the Stefan-Bolzmann constant $\sigma = 5.67 \times 10^{-8}$ W/m$^2$/K$^4$. Evaporation into a vacuum can be described by the Langmuir equation:

$$\frac{dm}{dt} = 4SCP_v\sqrt{\mu/T} \tag{2}$$

where $C = 4.377 \times 10^{-3}$, $\mu$ is molecular weight, $P_v$ is vapor pressure provided by the Antoine equation $P_v = A - B/T$. Finally, temperature was calculated from a nonlinear equation:

$$0.5\rho_a V^3 = 4(\sigma\varepsilon T^4 + H_v C P_v\sqrt{\mu/T}) \tag{3}$$

If vaporization is neglected, the temperature definition is straightforward[79]. However, such simplification leads to a substantial temperature overestimate. Although there is no obvious dependence for temperature on the particle size, the particle velocity dependence on altitude (and hence, the atmospheric density) is a function of the particle size. An emissivity of 0.95 is used in all simulations. Atmospheric density is defined by the ATMCIRA data. A list of constants used for the numeric modeling is provided in Table 1.

**Data availability**
The Fe isotopic composition and major and trace element composition of the Australasian macroscopic and microtektites data generated in this study have been deposited in the Mendeley data database (https://data.mendeley.com/) with the DOI identifier 10.17632/kmcpt5j7rz.1[81].
All numeric data is provided in tabular form in the electronic supplementary material. Source data are provided with this paper.

**Materials availability**
All Australasian tektites and microtektites are from collections of S.G., B.G., L.F. and M.G., as described in the materials section. Requests for samples should be sent to these authors. Smaller microtektites were nearly totally consumed for analysis in the process of this study. Source data are provided with this paper.

**Supplementary materials**
The Fe isotope ratio data together with a compilation of major and trace element concentrations measured in this work and in earlier studies[14–17,45,82,83] are provided in tabular form in the electronic supplementary material. Source data are provided with this paper.

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

## Acknowledgements

S.M.C. acknowledges his postdoctoral fellowship from the Flemish Research Foundation (FWO) in form of an "Excellence of Science (EoS)" project (ET-HoME - ID 30442502). C.G.dV. acknowledges the grant from Marie Curie Clarin-COFUND project of the Principality of Asturias and the European Union. B.S. and E.B.F. acknowledge their mandate grants from FWO. Collection of microtektites from Sør Rondane Mountains was made possible by the 2009 Baillet Latour Antarctica Fellowship to S.G.; S.G. and P.C. acknowledge the support from the FWO, BELSPO and the VUB Strategic Research. The FWO is acknowledged for providing the funding for the acquisition of the MC-ICP-MS instrumentation (ZW15-02 – G0H6216N). F.V. acknowledges the support from FWO under the form of the aforementioned EoS project and BOF-UGent. F.V. also acknowledges Teledyne CETAC Technologies for logistic and financial support. The TAM microtektite collection and research at Pisa University is supported by the Italian Programma Nazionale delle Ricerche in Antartide (PNRA), Meteoriti Antariche project (grant no. PNRA16_00029; PI. L.F.).

## Author contributions

S.M.Ch. and S.G. designed the concept of the research. S.M.Ch. wrote the initial manuscript, processed, analyzed and visualized the data, modeled the evaporation/condensation fractionation, and performed Fe isotopic and multi-elemental analyses. C.G.dV. performed Fe isotopic analysis and processed the raw isotope ratio data. N.A. performed numerical modeling and visualization of the impact event and the atmospheric entry, wrote part of the manuscript, and took part in conceptualization of the research. J.B. performed Fe isotopic analysis and analyzed the data concerning the weathering of glass. E.B.F. performed multi-elemental LA-ICP-MS analysis together with S.M.Ch. and took part in conceptualization of the research. L.F., B.G., M.G., M.V.G. and B.S. took part in sample selection, preparation, managed the microtektite collections, and took part in conceptualization of the research. F.V., P.C. and S.G. ensured funding and reviewed the manuscript. F.V. supervised the measurements at A&MS-UGent. S.G. supervised the project, took part in sample selection, data processing, visualization, analysis, conceptualization, and manuscript writing. All authors contributed to the discussion and interpretation of the results and proofread the manuscript.

## Competing interests

The authors declare no competing interests.
