## [Peer Review File · Nature Communications]

REVIEWERS' COMMENTS

Reviewer #1 (Remarks to the Author):

Reviewer: Andrew M. Davis

General Comments

This paper reports the Fe isotopic compositions of microtektites collected at a variety of distances from the likely impact location that generated the Australasian tektites. Interestingly, both isotopically light and heavy Fe have been found, with the light Fe near the impact site and heavy Fe further away. This is an important observation. The data appear to be robust and the conclusions, for the most part, are reasonable.

I find the section on implications for Solar System processes (L395–428) to be overly speculative, as conditions on Earth, with its highly oxidizing and significant atmosphere, are quite different from those on other bodies on which energetic collisions occur (asteroids, the Moon, the other terrestrial planets). The details of how the Fe isotopic fractionation occurred in microtektites are insufficiently well understood to regard these implications as anything other than pure speculation.

I recommend publication, but only after some revision to tone down the speculation a bit or at least add some caveats. There are a number of specific comments below that need to be addressed.

Specific Comments

- L60 References should be numbered in the order introduced in the text.
- L58 How is it known that microtektites are quenched in the upper atmosphere? I would expect that an object as small as a microtektite would quench by radiative cooling in space during its ballistic trajectory.
- L68 mineral unmelted fragments \Rightarrow unmelted mineral fragments
- L81 Wood et al. (2019, *Amer. Mineral.* **104**, 844–856) recalculated the condensation temperatures of the elements, taking particular care with solid solution models. Their condensation temperature for Fe, 1338 K, is not much different from Lodders (2003), but those of a number of volatile elements change by hundreds of degrees.
- L87–90 This has been demonstrated for Fe for evaporation of iron oxide and for iron-bearing silicate melts (Dauphas et al., 2004, ref. 44).
- L125 and \Rightarrow to
- L164–166 While it is true that evaporation is expected to lead to a linear array in the way described in this sentence, it seems odd not to show such a plot. Fig. 4 is not such a plot, being instead a plot of $\delta^{56/54}\text{Fe}$ vs. wt% FeO. Fig. 4 shows a scatter of data points with no hint of a linear array. This is true not only for all the microtektites, but also the SRM ones. Plotting $1000 \ln((^{56}\text{Fe}/^{54}\text{Fe})/(^{56}\text{Fe}/^{54}\text{Fe})_0)$ vs. $-\ln(\text{FeO}/\text{FeO}_0)$ would not magically make the data lie on a linear array.

- L205–209 If significant mass loss occurred during atmospheric reentry, I would expect significant evaporative Fe loss. This evaporative mass loss would under highly oxidizing conditions. Under these conditions, Ce becomes much more volatile than the other REE and has a volatility comparable to that of Fe (Wang et al., 2001, *Geochim. Cosmochim. Acta* **65**, 479–494). Fig. 2 shows no hint of Ce anomalies.
- L258–260 A kinetically controlled, or Rayleigh, condensation process should initially lead to isotopically light condensates, causing the vapor to evolve towards isotopically heavy compositions. See Richter (2004, ref. 37) for details on why. The isotopically light microtektites could be early condensates from a vapor plume.
- L263 First, the evaporating species from a silicate melt should be Fe_(g), not FeO_(g). Second, Dauphas et al. (2004) showed experimentally that Fe isotopic fractionation factor for evaporation from silicate melts is somewhat less than the ideal value.
- L301 Measured isotopic fractionation factors for iron evaporation (Dauphas et al., 2004) are much larger (further from 1) than the values here, indicating a more complex process.
- L355-357 Laboratory vacuum evaporation of FeO yields the ideal fractionation factor; evaporation of Fe from silicate is somewhat less fractionated than the ideal value (Dauphas et al., 2004).
- L370–373 I agree that ideal fractionation factors are hard to achieve in nature, but I don't think one can generalize the formation of microtektites to all natural volatility fractionation regimes. The formation of tektites in terrestrial impact plumes is likely to be very different from evaporation in other scenarios, like chondrule formation or CAI evaporation. There seems to be a lot back-reaction in chondrule formation and much less in CAI evaporation. Microtektites are interesting in their own right, but may not tell us about other kinds of events.
- L402 These ⇒ Those
- L407–409 While it may be a reasonable suggestion that light and heavy Fe isotopic signatures can be produced in the scenarios in this sentence, the required measurements have not been made and it is only conjecture at this point: “thus” is too strong a word.
- L415–424 These suggestions are interesting, but it is unclear whether they apply beyond Earth. A major difference between Earth and asteroids, Mercury, or the Moon, is that Earth has an atmosphere. The role of the atmosphere in both the expansion of the impact plume and the reentry from ballistic trajectories has not been fully explored.
- L511 Stephan ⇒ Stefan
- L538 Journals seem to be going away from page ranges to articles number. Nonetheless, it is useful to know the size of papers. For this reference, I would write: *Prog. Earth Planet. Sci.* **6**, #66 (10 pp) (2019). I don't know the policy of *Nature Communications* on this.
- L551–553 Ref. 11 is a book chapter. Since the entire book is written by Glass and Simonson, the book should be referenced, not the chapter.
- L564–565 Ref. 16 is also a book chapter. Since the entire book is written by Johnson, Beard, and Weyer, the book should be referenced, not the chapter.
- L591 O. ⇒ O'D.

- L612 Use superscripts for isotope numbers.
- L647–651 Don't capitalize every word in the title of the paper.
- L654 Use a subscript for SiO₂.
- L658 Is this a book? The reference seems incomplete.
- L664–665 Don't capitalize every word in the title of the paper.
- Table S2 Some values for MgO and Al₂O₃ seem to have too many significant figures (all zeros).

Reviewer #2 (Remarks to the Author):

This new study by Chernozhkin and coauthors reports LA-MC-ICP-MS Fe isotope data for tektites and microtektites collected from various locations. They found most Antarctic microtektites are enriched in heavy Fe isotopes, which has been attributed to evaporation that occurred either 1) in an impact plume or 2) during atmospheric reentry. They also found microtektites from some sediment cores are enriched in extremely light Fe isotopes. They excluded the possibilities of source heterogeneity and seawater alteration, instead proposing that these light Fe isotope signatures are caused by partial recondensation from the impact plume. This new data seems very interesting is the first time that Fe isotopes of microtektites have been reported. The Fe isotopic variation of microtektites is very significant and very different from the near-zero Fe isotope composition of all macroscopic tektites. They suggest that the processes forming tektites are much more convoluted than previously thought. The residual materials after impacts (e.g., tektites or lunar rocks) are not necessarily heavy in isotopic composition while the "lost" reservoir (e.g., impact removed early crust) are not necessarily light.

Overall, I think the interpretation is reasonable and I don't have any major issues. I think it is suitable for publication in Nature Communication after one round of revision clarifying the concerns I have listed below:

L79, what are the typical ranges of Fe isotopes of terrestrial igneous and sedimentary rocks? How narrow is the range? Some values and citations need to be added here to assist readers in comparing with the CI value provided. Why even mention the value for CI anyway, as the source rocks of tektites are clearly not CI meteorites?

L81, the nebular 50% condensation temperature may not be very relevant here. You can use Paolo Sossi's 1% evaporation temperature for Fe (2830K) in air at 1 bar, which is much more relevant to tektite formation conditions (Sossi et al., 2019, *Geochimica et Cosmochimica Acta*, Volume 260).

L91, what does this range of temperature refer to? Where is the temperature range coming from?

L95, is there any previous literature available on Fe isotopes of macroscopic tektites? There is no citation here.

L136, one additional figure plotting Fe isotopes vs. sample sizes would be useful. It would be easier to see than the color coded figure here.

L164, why only pick SRM here? How about other Antarctic microtektites (e.g., Larkman Nunatak)? They show an opposite trend (if any trend at all).

L165, how about using a ratio (FeO/Al₂O₃ or FeO/TiO₂ in Figure 7), rather than using the absolute concentrations? The absolute FeO concentrations can be affected by simply losing other volatiles without actually losing Fe.

L395, I am surprised that there are no discussion and comparison with chondrules. Previous studies have shown chondrules have either light or heavy Fe isotopic compositions, which are similar in range to those seen in microtektites (Hezel et al., 2018, *Earth and Planetary Science Letters*, Volume 490; Mullane et al., 2005, *Earth and Planetary Science Letters*, Volume 239; Hezel et al., 2010, *Earth and Planetary Science Letters*, Volume 296). The Fe isotopic variations among chondrules have been attributed to evaporation and recondensation, which is not too different from the interpretation here for microtektites.

L459, are BHVO and BCR matrix-matched to microtektites? I am asking because "coincidentally", those high-Mg ones (eg., 137-12, 137-8, 137-7: 15.88-18.52% MgO) are those showing the lightest in Fe isotopes when compared to those low-Mg ones in the same location (137-11, 137-10, 137-9: 2.18-3.59% MgO). Could large matrix difference in Mg cause matrix effect during analysis?

L462, at least the laser spot size needs to be reported here. How does the spot size compare to the size of microtektites? How many measurements have been done for each sample (please add number of measurements in Table S2)? Can multiple spots (from core to rim) of the same microtektite be measured? Are microtektites homogeneous in terms of Fe isotopes? I am curious to know whether there is any Fe isotope diffusion effect within the sphere, similar to what have been reported in Sio et al. (2013, *Geochimica et Cosmochimica Acta* volume 123, 302–321).

L745, typo: panes should be panels

L760, B, C, and D are three panels, one is for 0.5 s, one is for 5 s, what about the third one?

Supplementary material, the numbering of the literature in Table S1 does not seem to be correct.

Reviewer #3 (Remarks to the Author):

Dear Authors,

I appreciate the opportunity to review your manuscript. This work is very interesting and is of interest to geochemists who study impacts and volatile-depletion of planetary bodies, as well as impact modelers.

The major claims of this work are that microtektites can be reliably measured for Fe-isotopes that reflect processing during a hypervelocity impact. Those most distal to the impact event have the heaviest Fe isotopic signatures, and may also include the effects of atmospheric re-entry. This work is supported by modeling of impact melting and ejection of particles.

While this work is very interesting, it would strongly benefit from greater contextualization of existing work, both in the terrestrial and planetary science communities. In particular, I would start with Poitrasson (2004) on Fe isotopes in planetary bodies, as well as some of the later work that refers to these earlier measurements. More recent work on Fe isotopes in lunar samples by Day et al. (2019) may also be relevant.

Some of the statements made about evaporative fractionation and the formation of condensates during impact events are well understood or at least hypothesized when discussing isotopic measurements of volatile / moderately-volatile elements in tektites and planetary materials. More referencing of earlier work and contextualizing your study with this previous data would allow you to focus your paper on your results and model.

Additionally, it is important to clearly state that your study includes in situ measurements using laser ablation. How do those compare with previous measurements of Fe (or other volatile / moderately volatile) elements in planetary materials? (i.e. many of these are bulk analyses using wet chemical digestion techniques).

I think that this is novel, relevant and highly interesting work that would advance our understanding of impact events and their collateral effects. The main way to highlight this for this publication would be condense the text, refer to others for more basic concepts, and really focus on the contributions of your data and model -- this combination represents integrated and important work.

All the Best.

Line 42-47: consider adding a few general references with regard to planetary evolution

Line 47-51: The sentence is a bit confusing, perhaps consider something like the following
“As the isotope ratios of the moderately volatile elements are sensitive to evaporation and condensation, heavy isotopic compositions of these elements are observed for the following volatile-depleted bodies: the Moon, which formed as a result of ...

Line 52-53: Consider rewording a bit, so that it flows more:

“Terrestrial impact ejecta that formed during hypervelocity impacts that were prevalent in the early Solar System are especially interesting for studying the effects of evaporation that occurred during these events.”

Line 53-56: Consider switching sentence order

Line 60: consider rewording + start new paragraph here

“A key feature of tektites is their volatile-depletion, observed for water as well as other volatile and moderately volatile elements”

Line 62:

“tektite-microtektite” or “tektite and microtektite”

Line 63-66:

remove reference to Muong Nong type tektites, and rather explain it in the following sentence.

Line 63-69: please clarify

can be written more clearly to explain the logic behind why the Australasian tektites are linked to hypervelocity impact

*Also, I am not clear if Australasian and Muong Nong-type tektites are considered to be from the same impact....?

Line 71-73:

“as well” repeated in sentence

Line 76:

suggestion to remove “following” and replace “within”

Line 85:

perhaps consider adding more references after “debate” (to represent different points of view)

Line 85-90: consider moving into next paragraph, after first sentence.

Line 93-95: need references for isotope fractionation evidence of moon forming impact

Line 98-99: The delta values should be for one isotope (e.g., $\Delta^{66}\text{Zn}$ or $\Delta^{65}\text{Cu}$), and a generic formula can be provided.

Line 99:

change “correlated” to “that correlate”

Line 101-102: What element is this isotopic composition for? Please specify
Probably not necessary to specify per a.m.u, unless specific to referenced study. Per mil is acceptable notation.

Line 102: start new paragraph

Line 128: again, I'm not sure about the delta notation. It's typically one isotope (not the reference isotope)

Line 135-136:

Why is this "contrary to expectation"?

Perhaps this is because smaller microtektites have greater surface to volume ratio, and so surface condensates (typically lighter) are overrepresented?

Line 138:

"These data" or something – better not to write only "This", but rather specify so that the reader is clear on your meaning

Line 166:

"This [what]?" (please specify)

Line 179-181:

consider changing to: "fractionation mechanism of microtektites because they experience different temperature profiles over time, compared to their macroscopic analogues"

Also, why do they have more different temperature profiles compared to macroscopic tektites?

Line 184:

hint (subject is "signatures")

Line 187:

remove "progressively"

Line 201-213:

comparisons to atmospheric effects on micrometeorites may be relevant here

Line 219:

"isotopically light Fe component" is not sufficient. Need to explain how this would develop and be preserved in these tektites.

Line 227-256:

An overly detailed discussion of aspects that are not relevant to the light isotopic composition (condensation of vapor phase). Suggestion to condense this paragraph significantly.

Line 252:

need reference for LREE, Sr and Ba are mobilized easily

Line 259:

“evolve” (subject is “condensates”)

Line 258-260:

Confusing. If the light condensates are removed from the gas phase, it will evolve to isotopically heavier signatures. Important to be very clear in these mechanisms.

Line 257-283:

Paragraph would benefit from greater focus (and also condensing the text at the same time)

Line 282-283:

This should not be a final thought, but integrated much more into the discussion.

Line 285: I am unclear as to why FeO and Fe isotopic signatures should relate. Are there references and examples from previous studies? Are we assuming that all the Fe is FeO in the melt. That’s fine, but the assumption needs to be stated.

Line 297:

“starting material” is isotopically light? I don’t understand this, unless this is meant by the impactor. If so, are there examples of solar system materials with isotopically light Fe compositions? Assuming the target material is Earth composition, any heavy isotopes reflect evaporative fractionation and light isotopes indicate condensation of the resulting vapor phase (not starting material compositions).

Line 310:

change “displaying” to “that display”

Line 314-315:

Reference to lunar glass beads might be relevant here

Line 336:

I think that has also been suggested for other planetary bodies, although some of this work is ongoing and not completely published (but there may be abstracts)

Line 375:

change “different” to “In contrast”

Line 388: specify

“The latter scenario”

Line 374-394:

This discussion is very interesting and insightful. I would highlight it more, by removing some earlier discussion and focusing on such strong evidence of volatile depletion.

Line 397: remove slash, better to write it out

“evaporation or condensation”

Line 399:

“This relation”

Line 400-404:

GASP and HASP are defined twice

Line 404-407: This is ongoing work, there may be abstracts?

Line 411-412:

This is not necessarily correct. Condensates from the evaporated reservoir are often provided as an explanation for light isotopic compositions. Perhaps reference relevant works here.

Reviewer #1 (Remarks to the Author):

see attached review

Title: Isotopic evolution of the Solar System by hypervelocity impacts evidenced by Fe in microtektites

Authors: S. M. Chernozhkin, C. Gonzalez de Vega, N. Artemieva, B. Soens, J. Belza, E. Bolea-Fernandez, M. Van Ginneken, B. P. Glass, L. Folco, M. J. Genge, Ph. Claeys, F. Vanhaecke, S. Goderis

Reviewer: Andrew M. Davis

General Comments

This paper reports the Fe isotopic compositions of microtektites collected at a variety of distances from the likely impact location that generated the Australasian tektites. Interestingly, both isotopically light and heavy Fe have been found, with the light Fe near the impact site and heavy Fe further away. This is an important observation. The data appear to be robust and the conclusions, for the most part, are reasonable.

I find the section on implications for Solar System processes (L395–428) to be overly speculative, as conditions on Earth, with its highly oxidizing and significant atmosphere, are quite different from those on other bodies on which energetic collisions occur (asteroids, the Moon, the other terrestrial planets). The details of how the Fe isotopic fractionation occurred in microtektites are insufficiently well understood to regard these implications as anything other than pure speculation.

Dear prof. Davis,

First of all, we appreciate this highly detailed evaluation and the in-depth specific comments on our manuscript. The last part of the manuscript “implication for Solar System processes” was considerably shortened and re-written. The first part of the new paragraph focuses on lunar glasses, for which condensation in an impact plume was reported earlier. We also added more references, including the Herzog *et al.* Magnesium and Silicon Isotopes in HASP Glasses from Apollo 16 Lunar Soil 61241. LPSC, 43 1579 (2012). To the best of our knowledge, this is one of only very few reports on the isotopic signatures of HASP glasses. Although this represent only preliminary data, which was not followed by a peer-review publication, this report on HASP largely seems to be in agreement with our observations of condensed materials in terrestrial microtektites.

The second part of the paragraph was shortened significantly and now emphasizes that the observed Fe isotopic signatures and the suggested mechanism are much more complex than the previously suggested mechanism of isotope fractionation for hypervelocity impacts.

It is true that further constraints on the isotope fractionation mechanisms in the impact plume are needed to speculate on which effect they may have on the scale of planetesimals or planets. The suggestion that impact processes may have an effect contrary to expectations is now limited to a single sentence: “*Although the average Fe isotopic composition of the microtektites approximates that of the Bulk Silicate Earth, with a near-Gaussian frequency distribution (Fig.*

1), the possible observation of a spatial disconnection between the isotopically heavy evaporated and light condensed components (Fig. 3); may have planetary-scale implications”

Also, in several places in the text we added that the Earth’s atmosphere may represent a significant factor to induce isotope fractionation, which is known to differ in composition and volume from other planetary bodies.

I recommend publication, but only after some revision to tone down the speculation a bit or at least add some caveats. There are a number of specific comments below that need to be addressed.

Specific Comments

L50 References should be numbered in the order introduced in the text.

This error on numbering references was corrected in the text. To make sure this would not occur again, the Mendeley code was transformed to text before re-submission.

L58 How is it known that microtektites are quenched in the upper atmosphere? I would expect that an object as small as a microtektite would quench by radiative cooling in space during its ballistic trajectory.

This is a correct remark, a fraction of the microtektites may have quenched in space. But here the question is if all microtektites do reach space? As this is beyond the scope of this study and in order to avoid confusing the readers by the definition, we changed “quenched in the upper atmosphere” for more general “quenched in flight”.

L68 mineral unmelted fragments ⇒ unmelted mineral fragments

The requested correction was made in the text.

L81 Wood et al. (2019, Amer. Mineral. 104, 844–856) recalculated the condensation temperatures of the elements, taking particular care with solid solution models. Their condensation temperature for Fe, 1338 K, is not much different from Lodders (2003), but those of a number of volatile elements change by hundreds of degrees.

We have changed Lodders 2003 for the suggested reference. We appreciate the effort to update our manuscript with such up to date fundamental parameters. This sentence was re-written according to this comment and the comment of another reviewer: “*Iron is a comparatively refractory element (1338 K²⁰ 50% condensation temperature under nebular conditions, 10⁻⁴ atm, and 2830 K²¹ 1% evaporation temperature from silicate melts to air).*”

L87–90 This has been demonstrated for Fe for evaporation of iron oxide and for iron bearing silicate melts (Dauphas et al., 2004, ref. 44).

This reference was added to the sentence, as well as 2 others with a detailed description of the evaporation kinetics (as requested by another reviewer):

Davis, A. M. Volatile Evolution and Loss. in Meteorites and the Early Solar System II (eds. D.S. Lauretta; & H.Y. McSween Jr.) 295–307 (University of Arizona Press, 2006).

Richter, F. M. Timescales determining the degree of kinetic isotope fractionation by evaporation and condensation. Geochim. Cosmochim. Acta 68, 4971–4992 (2004).

L125 and \Rightarrow to

The requested correction was made in the text of the manuscript.

L164–166 While it is true that evaporation is expected to lead to a linear array in the way described in this sentence, it seems odd not to show such a plot. Fig. 4 is not such a plot, being instead a plot of $\delta^{56/54}\text{Fe}$ vs. wt% FeO. Fig. 4 shows a scatter of data points with no hint of a linear array. This is true not only for all the microtektites, but also the SRM ones. Plotting $1000 \ln((^{56}\text{Fe}/^{54}\text{Fe})/(^{56}\text{Fe}/^{54}\text{Fe})_0)$ vs. $-\ln(\text{FeO}/\text{FeO}_0)$ would not magically make the data lie on a linear array.

This is correct, the linearization in such coordinate space only exists for the V19-153 location, if compiled separately (we attached a figure of such a plot below and in the appendix). For the other locations, either the spread in the FeO concentration is not sufficiently large, or the data scatter is too high. We explain this scatter by admixture of variable proportions of the light condensed material or turbulence of the plume (both of which are inconsistent with the Rayleigh fractionation process).

While these sentences need to be rephrased (cf. below), the Rayleigh isotope fractionation process can be illustrated using either the “FeO- $\delta^{56}\text{Fe}$ ” or “ $\ln((^{56}\text{Fe}/^{54}\text{Fe})/(^{56}\text{Fe}/^{54}\text{Fe})_0) - -\ln(\text{FeO}/\text{FeO}_0)$ ” coordinate space. The practical use of the latter relative to the FeO- $\delta^{56}\text{Fe}$ is that the straight regression line can be fitted (using *e.g.* York’s model) and the slope can be used to calculate α with associated uncertainty. For this dataset, it is only practical for the V19-153 location for the reasons mentioned above. Even in this case, the calculation of α results in a relatively large uncertainty. The Rayleigh curves fitted “manually” in the dFe-FeO plots are as precise. Consequently, to avoid confusion we consider it better to keep the FeO- $\delta^{56}\text{Fe}$ coordinate space through the discussion, as it is more familiar for most readers. We rephrased the sentence in lines 164-166 and re-written it referring to the FeO- $\delta^{56}\text{Fe}$ coordinates (figure 4) as follows: “*The Fe isotope ratio data of the SRM microtektites are consistent with a system that follows a Rayleigh relationship with a starting composition characterized by $\delta^{56/54}\text{Fe} \approx 0$ and FeO concentration of 6.9 wt% (consistent with average crust), f equal to 0.71 – 0.58 and fractionation factor α varying between 0.995 and 0.997.*”

V19-153, E-N Indian Ocean

L205–209 If significant mass loss occurred during atmospheric reentry, I would expect significant evaporative Fe loss. This evaporative mass loss would under highly oxidizing conditions. Under these conditions, Ce becomes much more volatile than the other REE and has a volatility comparable to that of Fe (Wang et al., 2001, *Geochim. Cosmochim. Acta* 65, 479–494). Fig. 2 shows no hint of Ce anomalies.

We agree with the reviewer that this is what one would expect. For example, a fraction of cosmic spherules (melted micrometeorites) display negative Ce anomalies in their REE patterns (although this is not observed for all of them). As such, the absence of Ce* in microtektites compared to the presence of Ce anomalies in ~10-50% of cosmic spherules suggests that the evaporation during re-entry may be less significant. We added an extra sentence to point out this observation: “*Compared to most cosmic spherules, the mass loss experienced by the most distal Australasian microtektites during atmospheric re-entry appears to be significantly lower, as reflected by the absence of Ce anomalies in their REE patterns (Fig. 2). Cerium anomalies are occasionally observed in cosmic spherules as a result of oxidative conditions during evaporation*”⁴⁷.

“Occasionally” as an adverb here is due to the fact that not all cosmic spherules display Ce anomalies. The papers of Cordier (2011, 2012) reports about 50% CS with Ce anomalies, and they are even less frequent (~10% of all cosmic spherules) in the Antarctic collection from which the SRM microtektites were extracted (Goderis et al. *GCA* doi:10.1016/j.gca.2019.11.016). In our experience, Ce anomalies are only found in ~50% of the most strongly evaporated CAT and CAT-like cosmic spherules (“Elemental and oxygen isotopic analysis of highly vaporized cosmic spherules from Widerøefjellet, Sør Rondane mountains (East Antarctica), master dissertation of Tom Boonants, VUB 2020, unpublished work). It seems that the formation of such anomalies may depend on the initial presence of mineral phases, which can serve as reducing agents during the heating by gas drag.

L258–260 A kinetically controlled, or Rayleigh, condensation process should initially lead to isotopically light condensates, causing the vapor to evolve towards isotopically heavy compositions. See Richter (2004, ref. 37) for details on why. The isotopically light microtektites could be early condensates from a vapor plume.

We fully agree with the reviewer. However, although our results represent a significant step forward in understanding condensation processes in this context, we cannot resolve the exact underlying process at the current time. As such, the requested reference was added to the text, including a statement that different mechanisms leading to isotopically light condensates may be possible: “*Various mechanisms following condensation of oversaturated gas also remain possible, as this process is highly sensitive to pressure and cooling rate*³¹.”

Following the request of another reviewer, we have re-written the explanation of some potential mechanisms responsible for light condensate, focusing mainly on the classical explanation of Dansgaard 1964, given the complexity and diversity of more evolved condensation models, which are beyond the scope of this paper.

L263 First, the evaporating species from a silicate melt should be Fe(g), not FeO(g). Second, Dauphas et al. (2004) showed experimentally that Fe isotopic fractionation factor for evaporation from silicate melts is somewhat less than the ideal value.

Agreed. This sentence was deleted in the course of re-writing this paragraph (see the answer to the previous comment). It is true that the discussion of the exact pathway and specific mechanism of isotope fractionation in the plume should not be based purely on laboratory observations or theoretical predictions. As it is mentioned above, even under laboratory conditions the experimental α measured by Dauphas et al. 2004 for the evaporation from silicate melt of solar composition is $\sim 1.01322 \pm 67$, which is not what is expected for Rayleigh evaporation of Fe⁰. Yet, this $\alpha_{\text{experimental}}$ is suspiciously close to $\alpha_{\text{FeO}} = \text{SQRT}((56+16)/(54+16))=1.014$, although it might as well be that Fe⁰ is the evaporating species and the smaller fractionation is observed due to an effect of activity coefficients.

In the first version of the text, we used this purely theoretical calculation based on ideal parameters in L263 to demonstrate how incomplete evaporation followed by re-condensation can generate extremely light melts. Whether α_{FeO} or α_{Fe} is used for such calculation does not change the final composition by an order of magnitude and remains a purely theoretical approach to demonstrate the possibility to produce isotopically light melts. To avoid confusion and condense the text, we have deleted this sentence. A somewhat more elaborate discussion of the deviation of the observed α from the theoretical predictions is now included in the next section of the manuscript.

L301 Measured isotopic fractionation factors for iron evaporation (Dauphas et al., 2004) are much larger (further from 1) than the values here, indicating a more complex process.

This is a correct and valuable point. On line 375 (the section on non-ideal Rayleigh fractionation), we have expanded the text to read: “*All fractionation factors α fitted to the subsets of the microtektite data differ significantly from that predicted by ideal distillation to vacuum for both Fe (0.982) and FeO (0.986) species, suggesting a more complex mechanism.”*

Also, see answer to the next comment.

L355-357 Laboratory vacuum evaporation of FeO yields the ideal fractionation factor; evaporation of Fe from silicate is somewhat less fractionated than the ideal value (Dauphas et al., 2004).

Again, this comment is highly appreciated. On line 375, the very start of the section “non-ideal Rayleigh fractionation”, we have added a sentence to include the findings of Dauphas et al., 2004: “*Laboratory vacuum evaporation of FeO has previously constrained the ideal fractionation factor α , while evaporation of mixtures of oxides resulted in slightly less fractionated Fe isotope ratios than expected for the Rayleigh process*³².”

L370–373 I agree that ideal fractionation factors are hard to achieve in nature, but I don’t think one can generalize the formation of microtektites to all natural volatility fractionation regimes. The formation of tektites in terrestrial impact plumes is likely to be very different from evaporation in other scenarios, like chondrule formation or CAI evaporation. **There seems to be a lot back-reaction in chondrule formation and much less in CAI evaporation. Microtektites are interesting in their own right, but may not tell us about other kinds of events.**

Agreed. The sentence was corrected according to the comment, making this observation more specific to terrestrial impact processes only: “*As ideal Rayleigh fractionation factors α are rarely achieved in natural systems*^{33,35}, *the presented range of fractionation factors ($\alpha = 0.995$ to 0.999) fitted for Fe isotope fractionation in microtektites may be used as a first order estimate to interpret Fe isotope fractionation in terrestrial hypervelocity impact events.”*

Also, a short discussion of evaporation-condensation related isotope fractionation of chondrules and zoned metal in CB chondrites has been added to the “Solar System implications” section, following to the request of reviewer 2.

L402 These \Rightarrow Those

As this paragraph was re-written extensively according to the previous comments and the comments of the other reviewers, this syntax construction was changed entirely.

L407–409 While it may be a reasonable suggestion that light and heavy Fe isotopic signatures can be produced in the scenarios in this sentence, the required measurements have not been made and it is only conjecture at this point: “thus” is too strong a word.

We removed any mentioning of particular hypothetical scenarios when isotopically light condensates can be produced in the Solar System from this sentence. Instead, we now focus on the possibility to form such materials during impact events: “*Although it remains unclear under which conditions hypervelocity impacts produce isotopically light ejecta, the results of this work demonstrate that impact materials with Fe isotopic signatures both lighter and heavier than the target can be formed.*”

L415–424 These suggestions are interesting, but it is unclear whether they apply beyond Earth. A major difference between Earth and asteroids, Mercury, or the Moon, is that Earth has an atmosphere. The role of the atmosphere in both the expansion of the impact plume and the reentry from ballistic trajectories has not been fully explored.

Agreed. This part of the paragraph was significantly shortened to read: “*Although the average Fe isotopic composition of the microtektites approximates that of the Bulk Silicate Earth, with a*

near-Gaussian frequency distribution (Fig. 1), the possible observation of a spatial disconnection between the isotopically heavy evaporated and light condensed components (Fig. 3) may have planetary-scale implications. Whether condensation in the impact plumes affected the bulk compositions of bombarded bodies, or it remains a local feature of the microscopic spherules requires a deeper understanding of the mechanism of isotope fractionation in impact plumes and of the role of an atmosphere.”

L511 Stephan ⇒ Stefan

This mistake was corrected in the text.

L538 Journals seem to be going away from page ranges to articles number. Nonetheless, it is useful to know the size of papers. For this reference, I would write: *Prog. Earth Planet. Sci.* 6, #66 (10 pp) (2019). I don't know the policy of *Nature Communications* on this.

We added the number sign (#) whenever the article number is used instead of the page number in the references. However, we could not find any policy about this in the journal guidelines.

L551–553 Ref. 11 is a book chapter. Since the entire book is written by Glass and Simonson, the book should be referenced, not the chapter.

According to the comment, we changed this from citing the chapter to citing the entire book.

L564–565 Ref. 16 is also a book chapter. Since the entire book is written by Johnson, Beard, and Weyer, the book should be referenced, not the chapter.

Agreed and changed.

L591 O. ⇒ O'D.

The change was made in the text according to the comment.

L612 Use superscripts for isotope numbers.

This correction was made in the text.

L647–651 Don't capitalize every word in the title of the paper.

The text was changed according to the comment.

L654 Use a subscript for SiO₂.

This correction was implemented in the manuscript.

L658 Is this a book? The reference seems incomplete.

This reference is a report of Sandia National Laboratory. The bibliography list was updated to: “Thompson, S. L. & Lauson, H. S. Improvements in the CHART D radiation-hydrodynamic code III: revised analytic equations of state. *Sandia Natl. Lab. Rep.* **SC-RR-71 0** (1974).”

L664–665 Don't capitalize every word in the title of the paper.

This correction was made in the text.

Table S2 Some values for MgO and Al₂O₃ seem to have too many significant figures (all

zeros).

We reduced the number of significant figures where there were too many. Thanks for pointing this out.

Reviewer #2 (Remarks to the Author):

This new study by Chernozhkin and coauthors reports LA-MC-ICP-MS Fe isotope data for tektites and microtektites collected from various locations. They found most Antarctic microtektites are enriched in heavy Fe isotopes, which has been attributed to evaporation that occurred either 1) in an impact plume or 2) during atmospheric reentry. They also found microtektites from some sediment cores are enriched in extremely light Fe isotopes. They excluded the possibilities of source heterogeneity and seawater alteration, instead proposing that these light Fe isotope signatures are caused by partial recondensation from the impact plume. This new data seems very interesting is the first time that Fe isotopes of microtektites have been reported. The Fe isotopic variation of microtektites is very significant and very different from the near-zero Fe isotope composition of all macroscopic tektites. They suggest that the processes forming tektites are much more convoluted than previously thought. The residual materials after impacts (e.g., tektites or lunar rocks) are not necessarily heavy in isotopic composition while the “lost” reservoir (e.g., impact removed early crust) are not necessarily light.

Overall, I think the interpretation is reasonable and I don't have any major issues. I think it is suitable for publication in Nature Communication after one round of revision clarifying the concerns I have listed below:

We appreciate these comprehensive comments on our work and other suggestions on how to improve the manuscript. We answer the specific questions and suggestions point-by-point below.

L79, what are the typical ranges of Fe isotopes of terrestrial igneous and sedimentary rocks? How narrow is the range? Some values and citations need to be added here to assist readers in comparing with the CI value provided.

Excellent point. The description of the Fe isotopic composition typical for terrestrial rocks was improved. “*The Fe isotopic composition of Earth's mantle rocks ($\delta^{56/54}\text{Fe} = 0.002 \pm 0.224\%$, 2SD) is close to the Solar System average ($\delta^{56/54}\text{Fe} = -0.008 \pm 0.095\%$, based on 47 CI chondrites)¹⁸. The isotopic composition of Fe in the upper continental crust, as estimated from the composition of loess-paleosol sequences¹⁹, is slightly heavier ($\delta^{56/54}\text{Fe} = 0.09 \pm 0.03 \%$, 2SD). Terrestrial basalts have $\delta^{56/54}\text{Fe}$ of $0.085 \pm 0.130 \%$ (2 SD), while differentiated SiO₂ rich rocks display heavier Fe isotope values, in the range of 0.1 to 0.5 %¹⁸”.*

Why even mention the value for CI anyway, as the source rocks of tektites are clearly not CI meteorites?

We improved the description in this sentence, mentioning that the CI chondrites represent the best approximations for the initial composition of the Solar System.

L81, the nebular 50% condensation temperature may not be very relevant here. You can use Paolo Sossi's 1% evaporation temperature for Fe (2830K) in air at 1 bar, which is much more relevant to tektite formation conditions (Sossi et al., 2019, *Geochimica et Cosmochimica Acta*, Volume 260).

This reference was added to the text and the sentence was re-written: “*Iron is a comparatively refractory element (1338 K²⁰ 50% condensation temperature under nebular conditions, 10⁻⁴ atm and 2830 K²¹ 1% evaporation temperature from silicate melts to air).*”

L91, what does this range of temperature refer to? Where is the temperature range coming from?

This is a range of temperatures which ejecta experience during hypervelocity impact events. However, this sentence was re-written to condense the text and as a result any mentioning of the specific temperatures disappeared “*Isotope fractionation due to partial distillation during impact processes has a direct bearing on our understanding of key processes in the planetary sciences, such as asteroid or planetary collisions, for example during accretion or the Moon-forming impact*”³”.

L95, is there any previous literature available on Fe isotopes of macroscopic tektites? There is no citation here.

To the best of our knowledge, we are the first to report Fe isotopic compositions for macroscopic tektites. The word “*several*” was changed to “*other*” to emphasise this. The list of references (on the other isotope systems) was transferred to the end of this sentence “*Tektite glass represents a natural laboratory, which might be used to study impact-related Fe fractionation. The isotopes of other moderately volatile elements have previously been shown to fractionate in tektites as the result of extreme conditions during hypervelocity impacts*”^{4,33-40}”. The following sentences, which discuss each element separately, also contain these references.

L136, one additional figure plotting Fe isotopes vs. sample sizes would be useful. It would be easier to see than the color coded figure here.

An additional panel was added to figure 3 to illustrate the isotopic signatures of the Australasian microtektites versus their diameters. The figure caption was changed accordingly.

L164, why only pick SRM here? How about other Antarctic microtektites (e.g., Larkman Nunatak)? They show an opposite trend (if any trend at all).

Following this comment, we expanded the discussion on the nature of Fe isotope ratios in LN microtektites in the following section “*Overprinting of the Fe isotope fractionation mechanisms*”. We chose to discuss the LN and SRM microtektites separately because we initially organized the discussion of the data based on the isotope fractionation mechanism responsible

for the observed Fe isotope ratios. This allows to have a smoother narrative and to move from simple and expected concepts of evaporation towards the less expected process of condensation and finally to the more complex cases of the signatures which likely resulted from overprinting of multiple processes.

The SRM and LN microtektites are indeed not comparable. While the SRM microtektites display a modest evaporation trend in FeO – dFe space with minor dispersion, LN microtektites exhibit more heavily dispersed signatures, which indicate overprinting by multiple processes. At the end of the section “*Overprinting of the Fe isotope fractionation mechanisms*” it is now written: “*Large scatter around the Rayleigh evaporation trend for microtektites from the V20-138, V19-153 and LN locations might be explained by individual microtektites incorporating variable proportions of condensate diluted in the evaporated melt. This admixture might have taken place when the ballistic paths of these microtektites passed through turbulent and heterogeneous parts of the condensing impact plume. The Antarctic microtektites with the heaviest Fe isotope signatures appear to have experienced the least interaction with this isotopically light Fe component and/or underwent an additional episode of evaporation during atmospheric re-entry. Among the Antarctic collections, the LN microtektites display the largest within-site variability in $\delta^{56/54}\text{Fe}$ and the lowest values, suggesting considerable overprinting of the isotope fractionation mechanisms.*”

Also, the sentence in L164 was completely re-written according to the comment of another reviewer.

L165, how about using a ratio (FeO/Al₂O₃ or FeO/TiO₂ in Figure 7), rather than using the absolute concentrations? The absolute FeO concentrations can be affected by simply losing other volatiles without actually losing Fe.

We thank the reviewer for this suggestion. We added a second sub-plot to the figure 7 with FeO normalized to the sum of refractory TiO₂ and Al₂O₃. Such normalization was applied earlier to describe the results of the experiments of Fe evaporation to vacuum from silicate melts (Dauphas 2004). In the idealized model of Rayleigh distillation, the X-axis of the plot should of course contain the fraction of ⁵⁴Fe isotope remaining in the melt, which is in practice often changed to FeO content for simplicity. It is true that FeO can be affected by passive enrichment, and in the particular scenario of near-ideal Rayleigh distillation normalization to a refractory element appears to work well (Dauphas 2004). However, the processes involved in the impact event are significantly more complex. Firstly, at high temperatures and high evaporation degrees, when the melt loses dozens of percent of FeO, not even the refractory elements can be assumed to remain unchanged, including Ti and Al. As a result, an additional uncertainty of relative volatilities of elements is added to the equation. Secondly, and perhaps most importantly, the refractory elements in a condensation process are heavily enriched in the condensate compared to the initial composition, which would be reflected in a shift along the normalized X-axis.

As can be observed on the updated figure 7, normalization of the Fe content to refractory component confirms the trends already observed, indicating a complex process taking place during the impact event.

Fig. 7. Left – Plot of $\delta^{56/54}\text{Fe}$ versus the alkali elements normalized by the refractory element Ti to illustrate the effect of evaporation and condensation on volatile elements in microtektites. The Australasian microtektites that experienced the hottest conditions in the impact plume display the lowest concentrations in the alkali elements, together with the highest magnitudes of $\delta^{56/54}\text{Fe}$, both in positive and negative direction. The Antarctic microtektites may have experienced additional evaporation during atmospheric re-entry. Right – Diagram of $\delta^{56/54}\text{Fe}$ versus FeO normalized to major refractory elements to demonstrate that the absolute FeO concentrations are affected by volatile loss of Fe rather than the volatilization of other, more volatile major elements.

L395, I am surprised that there are no discussion and comparison with chondrules. Previous studies have shown chondrules have either light or heavy Fe isotopic compositions, which are similar in range to those seen in microtektites (Hezel et al., 2018, Earth and Planetary Science Letters, Volume 490; Mullane et al., 2005, Earth and Planetary Science Letters, Volume 239; Hezel et al., 2010, Earth and Planetary Science Letters, Volume 296). The Fe isotopic variations among chondrules have been attributed to evaporation and recondensation, which is not too different from the interpretation here for microtektites.

Additional discussion of condensation in the Solar Nebula was added to the section Implication for Solar System process: “*Evaporation and condensation in the early Solar Nebula are known to play a significant role in the Fe isotope budgets of chondrules, which experienced an episode of evaporation and re-condensation and have a $\delta^{56/54}\text{Fe} - \text{FeO}$ pattern resembling that of microtektites (Fig. 4)^{60,61}. The effects of impact evaporation on Fe are more scarce, e.g. condensation from an impact plume thought to lead to zoned metal in CB chondrites³⁰..” Note however that reviewer 1 also preferred to restrict a direct comparison with solar system processes.*

L459, are BHVO and BCR matrix-matched to microtektites? I am asking because “coincidentally”, those high-Mg ones (eg., 137-12, 137-8, 137-7: 15.88-18.52% MgO) are those showing the lightest in Fe isotopes when compared to those low-Mg ones in the same location

(137-11, 137-10, 137-9: 2.18-3.59% MgO). Could large matrix difference in Mg cause matrix effect during analysis?

This is a valid concern and in order to provide more details on our Fe isotope ratio measurement protocol, we have significantly extended the methodological description in the method's section: *“Validation of the protocol was performed using MPI-DING and USGS glass reference materials and cosmic spherules, which have highly variable matrices and Fe contents. The results of LA-MC-ICP-MS were in full agreement with the Fe isotopic signatures found in literature and with the results obtained following acid digestion and MC-ICP-MS measurement, independent of matrix compositions and the magnitude of the Fe signatures. LA-MC-ICP-MS $\delta^{56/54}\text{Fe}$ values of the glass GRMs and cosmic spherules, when plotted versus their $\delta^{56/54}\text{Fe}$ values measured by acid digestion, fall on a 1:1 line (0.9561 ± 0.0080) with MSWD = 1.7⁴⁶. For the in-length description of the protocol and the validation of the method, the reader is referred to previous work⁴⁶.”*

Prior to starting this work, we validated the LA-MC-ICP-MS analysis method and the limits of its application, including the influence of matrix effects. A peer-review paper on the method development was published in Analytical Chemistry 2020 (10.1021/acs.analchem.9b04029).

In this earlier paper, we validated the isotope ratio measurement protocol using glass GRMs of various compositions (rhyolite, komatiite, basalt, andesite, diorite) by comparing Fe isotope ratios measured *in-situ* with those available in literature and those measured using traditional digestion, Fe isolation and pneumatic nebulization of the solutions. Finally, we measured Fe isotope ratios of multiple glassy cosmic spherules *in-situ* (which have highly different matrices with up to 30 wt% differences in MgO), extracted the spherules from the epoxy, digested them, isolated Fe from matrix using anion exchange chromatography, and measured the results in solution (pneumatic nebulization, PN). The *in-situ* and bulk results were identical within uncertainty. The figure below shows the PN versus LA data of this earlier paper falling on 1:1 line.

The particularly low values measured for the 3 indicated microtektites are likely random. At the early stages of this work, we looked for Mg-rich microtektites specifically, as these were thought to reflect highly evaporated endmember compositions, but to provide conclusive evidence of

lighter isotopic signatures, a larger population is needed, preferably from the same location and of similar size. In total, we measured only 8 high-Mg/transitional microtektites, and this number is not sufficient to statistically say that they are lighter than the “normal” spherules. While the 197-1 is high-Mg microtektite (10.12 wt%, dFe = -0.50‰), microtektite 463-5 (6.41 wt% MgO) from the same location displays a significantly lower dFe of -1.86 ‰. Similarly, Antarctic high-Mg microtektites 4.6, 20.3, and 20.14 display near zero and heavy dFe signatures of -0.24, +1.05 and +1.13 ‰, respectively.

L462, at least the laser spot size needs to be reported here. How does the spot size compare to the size of microtektites? How many measurements have been done for each sample (please add number of measurements in Table S2)?

The information on the spot size, number of measurements and the laser frequency has been added to Table S2. The laser spot size and frequency were adjusted for each analysis in order to match the intensities of the Fe isotope signals between samples and standards within 3 to 5%. The laser energy and the dosage (number of shots per position) was always kept constant.

Can multiple spots (from core to rim) of the same microtektite be measured? Are microtektites homogeneous in terms of Fe isotopes? I am curious to know whether there is any Fe isotope diffusion effect within the sphere, similar to what have been reported in Sio et al. (2013, *Geochimica et Cosmochimica Acta* volume 123, 302–321).

It would be interesting indeed to investigate the kinetic effects inside the microtektites for potential condensation or diffusion due to cooling or Soret effect, but it is technically not possible at present. SIMS may have better spatial resolution, but it is also less precise than LA-MC-ICP-MS.

Spot analysis cannot be used for these analyses, in our setup we use line scans. First, a laser spot size of ~10-30 μm is needed for obtaining reasonable intensities of Fe isotopes. During 30 seconds of ablation, spot analysis would result in pits with high depth-to-diameter ratios, with significant signal suppression and in-depth artificial isotope fractionation. Potential mathematical correction is useless in this case because of the need to change the spot size between the samples and standards in order to fit the signal intensities of Fe isotopes within 3-5%. Because of this, the depth-to-diameters aspect ratios would vary for different pits. As such, line scans are more suited for in-situ Fe isotopic analysis. Unfortunately, this approach reduces the spatial resolution of the method. It is still possible to place adjacent lines to measure isotope profiles, but it is not practical for these tiny microtektites (mostly < 500 μm), as the lines are typically longer than the diameter of the spherules. On practice, we placed bent lines or “short rasters” across the spherules.

L745, typo: panes should be panels

Panes was changed for plots: “Upper plots are for a radius particle of 300 μm , bottom ones for a radius particle of 150 μm .”

L760, B, C, and D are three panels, one is for 0.5 s, one is for 5 s, what about the third one?

According to this comment, we added additional details in the caption of the figure 6: “(B, C, D) –Temperature cross-sections of the plume through the plane of symmetry XZ at time frames of 0.5, 1.5 and 4.6 seconds following the impact at 45 degrees”.

Supplementary material, the numbering of the literature in Table S1 does not seem to be correct.

We appreciate the careful reading through the manuscript by the reviewer - thanks. There was indeed a numbering error, which is now corrected.

Reviewer #3 (Remarks to the Author):

Dear Authors,

I appreciate the opportunity to review your manuscript. This work is very interesting and is of interest to geochemists who study impacts and volatile-depletion of planetary bodies, as well as impact modelers.

The major claims of this work are that microtektites can be reliably measured for Fe-isotopes that reflect processing during a hypervelocity impact. Those most distal to the impact event have the heaviest Fe isotopic signatures, and may also include the effects of atmospheric re-entry. This work is supported by modeling of impact melting and ejection of particles.

First of all, we appreciate these warm comments about our work and acknowledge the considerable amount of work done by the reviewer. The specific requests and comments are answered below point-by-point.

While this work is very interesting, it would strongly benefit from greater contextualization of existing work, both in the terrestrial and planetary science communities. In particular, I would start with Poitrasson (2004) on Fe isotopes in planetary bodies, as well as some of the later work that refers to these earlier measurements.

We appreciate the reviewer pointing this out. In order to highlight the significance of Fe isotopic composition as a proxy of planetary evolution, we added a brief literature overview in the introduction: *“Since the early reports on Fe isotopic compositions of the rocky bodies in the inner Solar System, it has been suggested that the observed differences between the Moon, Earth, Mars and some asteroids may result from variable degrees of evaporation experienced during their evolution.²² Although since then, it has become clear that the Fe isotopic composition of planetary surfaces is affected by both core-mantle segregation^{23,24} and redox-dependent²⁵ mantle differentiation,^{26,27} these processes alone may not be sufficient to account for the observed signatures of meteorites²⁸. As such, evaporation remains a viable process that could affect the Fe isotope budgets during planetary evolution either at the nebular stage, during the accretion, or subsequently following episodes of hypervelocity impacts or magmatism.^{28,29} Fractionation of Fe isotopes in impact plumes was observed earlier in the case of metal grains of CB chondrites³⁰.”*

In addition, in the Solar System implication section, we added a short comparison of our data with Fe isotope fractionation by condensation described earlier for chondrites: *“Evaporation and condensation in the early Solar Nebula are known to play a significant role in the Fe isotope budgets of chondrules, which experienced an episode of evaporation and re-condensation and have a $\delta^{56/54}\text{Fe} - \text{FeO}$ pattern resembling that of microtektites (Fig. 4)^{60,61}. The effects of impact evaporation on Fe are more scarce, e.g. condensation from an impact plume thought to lead to zoned metal in CB chondrites³⁰.”*

More recent work on Fe isotopes in lunar samples by Day et al. (2019) may also be relevant.

A brief discussion of the data of Day et al. 2019 [64] has been added in the section *“Implications for Solar System processes”*, where we describe the potential isotopic composition of evaporated and condensed reservoirs on the Moon: *“...Although no Fe isotope fractionation due to condensation was observed in bulk lunar breccia⁶⁴, GASP and HASP phases are likely to be*

characterized by strongly fractionated isotopic ratios for a range of elements based on their geochemical compositions⁶⁵...”

Some of the statements made about evaporative fractionation and the formation of condensates during impact events are well understood or at least hypothesized when discussing isotopic measurements of volatile / moderately-volatile elements in tektites and planetary materials. More referencing of earlier work and contextualizing your study with this previous data would allow you to focus your paper on your results and model.

The first part of the the “Implication for Solar System processes” was re-written significantly according to the comments of the reviewers, and more earlier works on evaporation and condensation in the planetary materials are discussed to contextualize the findings of this work.

Additionally, it is important to clearly state that your study includes *in situ* measurements using laser ablation. How do those compare with previous measurements of Fe (or other volatile / moderately volatile) elements in planetary materials? (i.e. many of these are bulk analyses using wet chemical digestion techniques).

We agree with the reviewer. To clearly point out that our measurements were done *in situ*, we added a statement in the goals section: “*To determine the Fe isotopic compositions of these microscopic glasses in situ, we used laser ablation coupled to a multicollector ICP mass spectrometer.*”.

Validation of the analytical protocol was an important part of the work, which was performed using a set of USGS and MPI-DING GRMs with glassy matrix. The $\delta^{56}\text{Fe}$ and $\delta^{57}\text{Fe}$ measured *via* LA-MC-ICP-MS, taking into account the uncertainty, were in full agreement with those obtained with the traditional method that consists of acid digestion and Fe anion exchange isolation followed by pneumatic nebulization MC-ICP-MS analysis. Also, the *in situ* results were in good agreement with the literature values for these GRMs. Finally, we observed excellent agreement between the Fe isotopic signatures of several glassy cosmic spherules (micrometeorites) measured *in situ* and *via* the pneumatic nebulization MC-ICP-MS. After the LA-MC-ICP-MS analysis, a subset of spherules was extracted from the epoxy mount and their measured “bulk” signatures were found to be in good agreement with the *in situ* data. The figure below shows that the solution data (PN for pneumatic nebulization) when plotted versus LA data, fall onto 1:1 line within uncertainty (MSWD = 1.7)

The complete validation of the LA-MC-ICP-MS measurement protocol of Fe isotopic analysis of glassy matrices was presented in the earlier peer-reviewed publication (Gonzales de Vega et al. *AnalChem* 2020, Vol 92 (5), 3572-3580, DOI: 10.1021/acs.analchem.9b04029). This manuscript also contains a detailed description of the protocol. In order to ensure the reader that the results of LA-MC-ICP-MS analysis are in line with more traditional approach, we modified the way we refer to our earlier paper in the methods section: “*The analysis was performed by ablating along a line with a laser spot size of 10-30 μm and a laser repetition rate of 10-30 Hz. Validation of the protocol was performed using MPI-DING and USGS glass reference materials and cosmic spherules, which have highly variable matrices and Fe contents. The results of LA-MC-ICP-MS were in full agreement with the Fe isotopic signatures found in literature and with the results obtained following acid digestion, chromatographic Fe isolation and MC-ICP-MS measurement, independent of matrix compositions and the magnitude of the Fe signatures. LA-MC-ICP-MS $\delta^{56/54}\text{Fe}$ values of the glass GRMs and cosmic spherules, when plotted versus their $\delta^{56/54}\text{Fe}$ values measured after acid digestion, fall on a 1:1 line (0.9561 ± 0.0080) with MSWD = 1.7⁴⁶. For the in-length description of the protocol and the validation of the method, the reader is referred to previous work⁴⁶.*”

I think that this is novel, relevant and highly interesting work that would advance our understanding of impact events and their collateral effects. The main way to highlight this for this publication would be condense the text, refer to others for more basic concepts, and really focus on the contributions of your data and model - this combination represents integrated and important work.

The text was re-written substantially according to the comments of all reviewers to improve the overall readability and to trim out the less significant parts. An entire section of the text (Potential reasons for isotopically light Fe in microtektites, other than condensation) was moved to the appendix, and only a shortened version of this paragraph is left in the main text. The description of the important aspects of work and mechanisms of fractionation was refined, and the section of implication for Solar System materials was re-written significantly to focus on the results of this work rather than on hypothetical scenarios.

All the Best.

Line 42-47: consider adding a few general references with regard to planetary evolution

The citations of the following works were added to highlight the first 3 sentences of the introduction:

Davis, A. M. Volatile Evolution and Loss. in *Meteorites and the Early Solar System II* (eds. D.S. Lauretta; & H.Y. McSween Jr.) 295–307 (University of Arizona Press, 2006).

Keil, K., Stöffler, D., Love, S. G. & Scott, E. R. D. Constraints on the role of impact heating and melting in asteroids. *Meteorit. Planet. Sci.* **32**, 349–363 (1997)

Line 47-51: The sentence is a bit confusing, perhaps consider something like the following “As the isotope ratios of the moderately volatile elements are sensitive to evaporation and condensation, heavy isotopic compositions of these elements are observed for the following volatile depleted bodies: the Moon, which formed as a result of ...

The sentence was changed according to this comment: “*The isotope ratios of the moderately volatile elements are sensitive to evaporation and condensation. Heavy isotopic compositions of these elements are observed for the Moon³, formed as the result of a giant impact, the volatile-depleted parent asteroids of the ureilite⁴ and angrite⁵ meteorites, asteroid Vesta⁶, and potentially also for Mercury with its partially stripped mantle⁷.*”

Line 52-53: Consider rewording a bit, so that it flows more: “Terrestrial impact ejecta that formed during hypervelocity impacts that were prevalent in the early Solar System are especially interesting for studying the effects of evaporation that occurred during these events.”

The sentence was corrected according to the comment: “*Terrestrial ejecta, which formed during hypervelocity impacts that were especially prevalent during the early Solar System, are particularly interesting for studying the effects of evaporation that occurred during these events.*”

Line 53-56: Consider switching sentence order

The sentence was corrected: “*Such impacts generate ejected silicate melt with broadly upper crustal compositions as a result of extreme heat and pressure.*”

Line 60: consider rewording + start new paragraph here “A key feature of tektites is their volatile-depletion, observed for water as well as other volatile and moderately volatile elements”

The text was corrected according to this comment and a paragraph break was added: “*A key feature of tektites is their volatile-depletion, observed for water, as well as for volatile and moderately volatile elements⁹.*”

Line 62: “tektite-microtektite” or “tektite and microtektite”

The text was corrected according to the comment “tektite and microtektite”

Line 63-66: remove reference to Muong Nong type tektites, and rather explain it in the following sentence.

The reference to the Muong Nong type Australasian tektites was removed.

Line 63-69: please clarify. can be written more clearly to explain the logic behind why the Australasian tektites are linked to hypervelocity impact

Part of this paragraph was rewritten to provide more clarity for the reader:

“*Australasian tektites and microtektites are linked to a hypervelocity impact origin by their geochemical compositions and features characteristic for shock melting: the absence of primary minerals, the presence of lechatelierite and shocked mineral grains in some tektites and microtektites,¹⁰ as well as the occurrence of high-pressure mineral phases in some tektites¹¹. The recovery of shock metamorphosed rock and unmelted mineral fragments in microtektite layers in the South China Sea¹² further supports the impact origin.*”

*Also, I am not clear if Australasian and Muong Nong-type tektites are considered to be from the same impact....?

All references to the Muong Nong type tektites were removed, as it has secondary significance. Muong Nong refers to one of the textural types of the Australasian tektites.

Line 71-73: “as well” repeated in sentence

“as well as” was changed for “and”

Line 76: suggestion to remove “following” and replace “within”

The requested correction was introduced into the text: “*Iron is a major rock-forming element in most planetary reservoirs with variable geochemical behavior and an assortment of contexts.*”

Line 85: perhaps consider adding more references after “debate” (to represent different points of view)

An extra reference to Poitrasson et al. *Geochim. Cosmochim. Acta* **267**, 257–274 (2019) was added, propagating the effect of evaporation during accretion onto the Fe isotopic composition. An entire block of text was re-written to provide the reader with more context on Fe isotopes in planetary bodies, as requested in an earlier comment. This paragraph includes several extra citations of existing publications.

Line 85-90: consider moving into next paragraph, after first sentence.

The requested change was made to the text.

Line 93-95: need references for isotope fractionation evidence of moon forming impact.

A reference to the work observing potassium isotope fractionation between the Earth, Moon and chondrites was added to the text: Wang, K. & Jacobsen, S. B. Potassium isotopic evidence for a high-energy giant impact origin of the Moon. *Nature* **538**, 487–490 (2016).

Line 98-99: The delta values should be for one isotope (e.g., $\delta^{66}\text{Zn}$ or $\delta^{65}\text{Cu}$), and a generic formula can be provided.

An additional sentence was added to the methods section: “*The delta notations for Cu, Zn and Sn discussed in the text are calculated in a similar way as for Fe using the ^{63}Cu , ^{64}Zn and ^{118}Sn reference nuclides.*”

We did not find anything specific in the journal guidelines for reporting the isotope deltas, so unless the editor specifies otherwise, we prefer to keep the $\delta^{56/54}\text{Fe}$ notation, based on recommendations in publications of the Commission on Isotopic Abundances and Atomic Weights of the IUPAC, as described by Coplen 2011, *Rapid communications in mass spectrometry* **25** P. 2538-2560 (DOI: 10.1002/rcm.5129). As Fe has more than 2 isotopes, such notation avoid confusions and additional explanations.

Line 99: change “correlated” to “that correlate”

The corresponding correction was made to the text.

Line 101-102: What element is this isotopic composition for? Please specify. Probably not necessary to specify per a.m.u, unless specific to referenced study. Per mil is acceptable notation.

The sentence was re-phrased to improve the readability: “*At the same time, other volatile elements exhibit no isotope fractionation due to evaporation in macroscopic tektites. As an example, the isotopic variations of Li and K only reflect mixing of distinct reservoirs within an inhomogeneous target*^{34,38–40}.”

Line 102: start new paragraph

The corresponding change was made to the text.

Line 128: again, I'm not sure about the delta notation. It's typically one isotope (not the reference

isotope)

As noted in the earlier answer, we did not find anything specific in the journal guidelines for reporting the isotope deltas, so unless the editor specifies otherwise, we prefer to keep the $\delta^{56/54}\text{Fe}$ notation, based on recommendations in publications of the Commission on Isotopic Abundances and Atomic Weights of the IUPAC, as described by Coplen 2011, *Rapid communications in mass spectrometry* **25** P. 2538-2560 (DOI: 10.1002/rcm.5129). As Fe has more than 2 isotopes, such notation avoid confusions and additional explanations.

Line 135-136: Why is this “contrary to expectation”? Perhaps this is because smaller microtektites have greater surface to volume ratio, and so surface condensates (typically lighter) are overrepresented?

The reviewer is correct, but as this expression lead to confusion “contrary to expectations” was deleted. Our initial expectation was to see no variation at all in microtektites.

Line 138: “These data” or something – better not to write only “This”, but rather specify so that the reader is clear on your meaning

The corresponding correction was made to the text: “*These high contents may reflect a distinct endmember composition within the target lithologies or an immiscible Fe rich melt known to occur within specific tektites^{14,44}*”

Line 166: “This [what]?” (please specify)

The sentence was re-written entirely according to this comment and that of the first reviewer “...*The Fe isotope ratio data of the SRM microtektites are consistent with a system that follows a Rayleigh relationship with a starting composition characterized by $\delta^{56/54}\text{Fe}$ of ~ 0 and FeO concentration of 6.9 wt% (consistent with average crust), f equal to 0.71 – 0.58 and fractionation factor a between 0.995 and 0.997*”.

Line 179-181: consider changing to: “fractionation mechanism of microtektites because they experience different temperature profiles over time, compared to their macroscopic analogues”

Also, why do they have more different temperature profiles compared to macroscopic tektites?

A corresponding correction was made to the text: “*The microscopic size thus plays a critical role in the isotope fractionation mechanism of microtektites, because they experienced different temperature profiles over time, compared to their macroscopic analogues due to different surface-to-volume ratios.*”

Line 184: hint (subject is “signatures”)

Changed.

Line 187: remove “progressively”

This correction was made to the text.

Line 201-213: comparisons to atmospheric effects on micrometeorites may be relevant here

A comparison to cosmic spherules was added to the sentence: “*Different from the much faster cosmic spherules⁴⁶, gas drag cannot contribute to heating of the microtektites over most of their ballistic trajectories, as they escape the atmosphere.*”

Because some of the micrometeorites have velocity/mass that is insufficient for melting during atmospheric entry, cosmic spherules are probably a better comparison here than micrometeorites. Cosmic spherules are a textural sub-type of micrometeorites that indeed experienced strong heating due to gas drag and as a result have extreme variations in Fe isotopic composition, as shown in our earlier publication (cf. above).

Line 219: “isotopically light Fe component” is not sufficient. Need to explain how this would develop and be preserved in these tektites.

An additional detail was added to the sentence: “*Alternatively, TAM and LN microtektites could contain a higher proportion of an isotopically light Fe component, a product of condensation, relative to those from the SRM.*”

Line 227-256: An overly detailed discussion of aspects that are not relevant to the light isotopic composition (condensation of vapor phase). Suggestion to condense this paragraph significantly.

This paragraph was reduced significantly according to this request. However, because this discussion is important to rule out all other processes that can make the microtektites isotopically light (weathering, equilibration with water or an even initially isotopically light target), we prefer to keep the in-length version of this paragraph in the appendix.

Line 252: need reference for LREE, Sr and Ba are mobilized easily

The reference on weathering was added in the Appendix A, as now this block of text was transferred into the appendix following the previous comment: “*Pourkhorsandi, H. et al. The effects of terrestrial weathering on samarium-neodymium isotopic composition of ordinary chondrites. Chem. Geol. 562, 120056 (2021).*”

Line 259: “evolve” (subject is “condensates”)

We revised this overcomplicated sentence: “*For example, condensation of isotopically heavy droplets can progressively shift the gas phase towards the light isotopic signatures. Over time, the instant condensate can become isotopically lighter compared to the initial isotopic composition⁵².*”

Line 258-260: Confusing. If the light condensates are removed from the gas phase, it will evolve to isotopically heavier signatures. Important to be very clear in these mechanisms.

The description is revised for better clarity, please see the answer to the previous comment. As the condensation process is sensitive to pressure and cooling rates (on the time scale of the condensation), there are multiple mechanisms by which the condensate can become isotopically light. We find it difficult to explain in a simple way all of these complex kinetic models, but we provide an explanation for a selection of pathways. The classical explanation, used for enrichment of light isotopic species in meteoric water, is also valid here. As the heavy

condensate is removed for a period of time, the gas phase shifts towards the light values and starts to generate condensate lighter than the initial composition. But it will still be heavier than the gas phase at this certain moment. Additional references are added, which describe the kinetics of condensation and evaporation at length:

Richter, F. M. Timescales determining the degree of kinetic isotope fractionation by evaporation and condensation. *Geochim. Cosmochim. Acta* **68**, 4971–4992 (2004).

Davis, A. M. Volatile Evolution and Loss. in *Meteorites and the Early Solar System II* (eds. D.S. Lauretta; & H.Y. McSween Jr.) 295–307 (University of Arizona Press, 2006).

Line 257-283: Paragraph would benefit from greater focus (and also condensing the text at the same time)

The paragraph was re-written and the secondary ideas were shortened to improve focus on the critical idea, which is the presence of condensation process during the impact event.

Line 282-283: This should not be a final thought, but integrated much more into the discussion.

The sentence was expanded: “*In the case of macroscopic tektites, the effect of condensation is likely insignificant as a result of their low surface-to-volume ratios, and light isotopic signatures of Fe and K have not been observed to date.*”

Line 285: I am unclear as to why FeO and Fe isotopic signatures should relate. Are there references and examples from previous studies? Are we assuming that all the Fe is FeO in the melt. That’s fine, but the assumption needs to be stated.

In order to present the effect of Rayleigh evaporation, an explanation in layman’s terms was added in the introduction, and it was complemented with 2 additional references to specific literature where the kinetics of evaporation of isotopes are explained in detail: “*As a result of evaporation, the concentration of the moderately volatile elements in the residual phase is reduced and their isotopic composition shifts towards heavier values^{1,31,32}.*”

Line 297: “starting material” is isotopically light? I don’t understand this, unless this is meant by the impactor. If so, are there examples of solar system materials with isotopically light Fe compositions? Assuming the target material is Earth composition, any heavy isotopes reflect evaporative fractionation and light isotopes indicate condensation of the resulting vapor phase (not starting material compositions).

This is correct. The suggestion that the only explanation of the light “starting material” is through condensation is exactly what we meant to show in this overly confusing sentence. The idea is that the microtektites from these locations were formed by 1) condensation, which resulted in light signatures, that were followed by 2) evaporation, which resulted in a typical Rayleigh – like evaporation trend in FeO- δ Fe space. There are materials in the Solar System, which may be isotopically light (e.g., troilite), but their presence is not confirmed by the major element composition of the ejecta. This sentence is now corrected to: “*However, compared to the putative target composition, these trends suggest that the microtektites from V20-138 and V19-153 formed by evaporation from the material that initially had an extremely light Fe isotopic signature ($\delta^{56/54}\text{Fe}$ between -2.61 and -2.85‰) and an elevated FeO content between 7.19 and 7.32 wt%. Such initial compositions are in agreement with the lightest isotopic signatures preserved among the microtektites measured here, as a result of condensation.*”

Line 310: change “displaying” to “that display”

The requested correction was made to the text.

Line 314-315: Reference to lunar glass beads might be relevant here

A reference to the lunar glass beads was added to this sentence: “*Although evaporation during ballistic transport and interaction with the atmosphere cannot be excluded in the case of the most distally deposited (Antarctic) microtektites, evaporation must also take place earlier on in the impact plume, as suggested for lunar glass beads that form in the absence of an atmosphere.*”

Line 336: I think that has also been suggested for other planetary bodies, although some of this work is ongoing and not completely published (but there may be abstracts)

We thank the reviewer for this comment. A reference to an LPSC abstract suggesting a heterogeneous condensation mechanism in volcanic plumes for glassy spherules in Apollo samples has been added to the next sentence (Meyer et al. The source of sublimates on the Apollo 15 green and Apollo 17 orange glass samples, LPSC 1975). We also added a more recent report of heterogeneous condensation in an impact plume on the Moon (Mokhov et al. A High-Temperature Al–Ca Condensate in the Lunar Regolith. *Dokl. Earth Sci.* **487**, 815–818, 2019): “*Similarly, a thin layer of material covering the core inner part has been observed for lunar glasses, suggesting heterogeneous condensation in volcanic⁵⁴ and impact⁵⁵ plumes.*”

However, we could not find abstracts where a similar mechanism has been suggested for Solar System bodies other than the Earth or the Moon. There is indeed a paper (Wittmann et al., 2015 MAPS “Petrography and composition of Martian regolith breccia meteorite Northwest Africa 7475”) on Martian regolith NWA7475 that contains glassy impact spherules, but it reports no discussion about their potential condensation mechanism. As such, no references to unpublished work are included here. However, we added a reference to this manuscript in a later section when we discuss that the microtektite-style microscopic ejecta is not unique to the Earth in a dynamic Solar System.

Line 375: change “different” to “In contrast”

The requested change was made to the text.

Line 388: specify “The latter scenario”

The requested correction was made in the text. Now the sentence reads as follows: “*Overprinting of condensation and evaporation results in microtektites that are characterized by Fe isotopic signatures near the starting value but with strong depletions in the alkali elements.*”

Line 374-394: This discussion is very interesting and insightful. I would highlight it more, by removing some earlier discussion and focusing on such strong evidence of volatile depletion.

The earlier discussions were re-written significantly, and this paragraph is more in place now. We also improved the figure 7 according to the comment of the other reviewer.

Line 397: remove slash, better to write it out “evaporation or condensation”

The requested change was made in the text.

Line 399: “This relation”

The requested change was made in the text.

Line 400-404: GASP and HASP are defined twice

The confusing brackets (GASP) and (HASP) in the second sentence were deleted: “*These glassy spheroids are thought to have formed from lunar crustal materials as condensates and partial evaporates, respectively, within impact plumes on the Moon...*”

Line 404-407: This is ongoing work, there may be abstracts?

Unfortunately, we do not have access to Apollo or Luna materials, but it would be a logical next topic to address. We added a reference to the LPSC abstract of Herzog, G. F. *et al.* “Magnesium and Silicon Isotopes in HASP Glasses from Apollo 16 Lunar Soil 61241. in *Lunar and Planetary Science Conference, 43* 1579 (2012)”, who presented Si, Mg isotopic compositions for a few HASP glasses and impact spherules from the Apollo materials, measured using SIMS. The results demonstrate significant isotope fractionation towards lighter values, suggesting that condensation took place in the impact plume. But the authors admit that the results might also reflect an analytical artifact. Although this abstract is not followed up by a full paper, the preliminary results of Herzog *et al.* are fully consistent with the discussion of condensation in this manuscript.

Line 411-412: This is not necessarily correct. Condensates from the evaporated reservoir are often provided as an explanation for light isotopic compositions. Perhaps reference relevant works here.

We added a reference to a paper by Moynier *et al.* “Volatilization induced by impacts recorded in Zn isotope composition of ureilites. *Chem. Geol.* **276**, 374–379 (2010).”, in which heavy Zn isotopic compositions in the ureilite parent body are explained by impact evaporation.

REVIEWERS' COMMENTS

Reviewer #1 (Remarks to the Author):

The authors have rewritten their paper carefully taking into account the comments made by me and two other reviewers. I am satisfied with their reaction to my comments, where they either made changes to the manuscript or provided a reasonable rebuttal. I recommend publication.

Reviewer #2 (Remarks to the Author):

The authors have addressed all my concerns. I have no further issues and I am glad to recommend its publication.

Reviewer #3 (Remarks to the Author):

Dear Authors,

I recognize your careful incorporation of reviewer comments and suggestions into the manuscript, and appreciate that changes were explicitly marked. This is a much improved manuscript, and strengthens the impact of your meticulous and novel work.

In addition to the attached and minor line-by-line comments, I have few questions / points to consider.

1) How do the LA-ICP-MS analyses compare with secondary ion mass spectrometry (SIMS) measurements? I think this is a natural question that readers may ask. I understand that the microtektites are small, but I think generally SIMS spots are smaller than LA-ICP-MS. This only has to be a sentence, but I think that it should be addressed.

Relatedly, I greatly appreciate your explaining the methods in greater detail and responding to my earlier concerns.

The figure that you provided in the reviewer response may also be beneficial as a supplementary figure, if you think that would be appropriate and help other readers

2) In the conclusion, there is discussion of GASP and HASP, which are specific to the Moon. I think these sentences can be moved up earlier, as there are other places where the Moon and lunar glasses are mentioned. In the conclusion, perhaps provide a summary sentence of comparisons, rather than introducing an entirely new concept here

3) I find it interesting that chondrules have similar signatures. Though this may be beyond the scope of this work, is there any evidence that the conditions of chondrule formation may be similar to hypervelocity impacts?

4) Depending on publication limits, if need be, Figures 5-7 could be moved to the supplement.

6) For background, there are many references missing. To recognize that there are additional references, please use e.g. in these citations, for example in lines 49-51. There is no need to add references, especially given limits on space, but it is important to not cite references as if they are the definitive source for a statement.

All the Best!

Line 47-58: briefly describe heavy versus light isotopic signatures here. Could move lines 104-108 here

Line 55-58: switch sentences, so that it reads: "Such impacts...Tektites..."

Line 71: Please clarify, i.e. I think the South China Sea is part of the Australasian strewn field, but I am not sure?

Line 140-141: Are petrographic observations from this study? If so, what are the signs of alteration? If not, then please reference the appropriate study.

Line 160: Please use "e.g." to describe moderately volatile elements because there are more than are mentioned in this line.

Line 196: change "hint towards" to "indicate"

Line 314-315: reference needed for lunar glasses

Line 282: This heading is a little unclear and vague, "Overprinting of the Fe isotope fractionation mechanisms," The overprinting is by condensation, and should be specified as such. Additionally, the vapor phase that condenses is part of the Fe isotope fractionation mechanism. Perhaps "Overprinting of heavy Fe isotope signatures by light vapor condensate"?

Line 368: Is it possible to quantify these statements? "move fast" / "very quickly"

Line 389-391: This scenario could be compared to differences in Cu versus Fe condensation, see Day et al., (2019)

Reviewer #1 (Remarks to the Author):

The authors have rewritten their paper carefully taking into account the comments made by me and two other reviewers. I am satisfied with their reaction to my comments, where they either made changes to the manuscript or provided a reasonable rebuttal. I recommend publication.

Dear prof. Davis,

We acknowledge your evaluation of and comments on our manuscript, and we are happy to read that our corrections covered the points indicated and changed the focus of the paper in the right direction.

Sincerely,

Stepan M. Chernozhkin and co-authors

Reviewer #2 (Remarks to the Author):

The authors have addressed all my concerns. I have no further issues and I am glad to recommend its publication.

Dear anonymous reviewer,

Once again, we acknowledge these comments and suggestions provided to improve our work.

Sincerely,

Stepan Chernozhkin and co-authors

Reviewer #3 (Remarks to the Author):

Dear Authors,

I recognize your careful incorporation of reviewer comments and suggestions into the manuscript, and appreciate that changes were explicitly marked. This is a much improved manuscript, and strengthens the impact of your meticulous and novel work.

In addition to the attached and minor line-by-line comments, I have few questions / points to consider.

Dear anonymous reviewer,

First, we appreciate the extra mile went to help us improve the quality of the manuscript. The specific requests and comments are answered below point-by-point.

1) How do the LA-ICP-MS analyses compare with secondary ion mass spectrometry (SIMS) measurements? I think this is a natural question that readers may ask. I understand that the microtektites are small, but I think generally SIMS spots are smaller than LA-ICP-MS. This only has to be a sentence, but I think that it should be addressed.

This is a valid request. Fe isotopic analysis using SIMS is indeed generally characterized by smaller spot size, down to 15-20 μm [e.g. Marin-Carbonne et al. 2011 Chem. Geol. 285, 50–61. DOI:10.1016/j.chemgeo.2011.02.019], at nearly similar precision of isotope ratio measurements: i.e. 0.2–0.3‰ for SIMS versus 0.04–0.22 for nsLA-MC-ICP-MS (2σ for $\delta^{56}\text{Fe}$, Marin-Carbonne et al. 2011 and Gonzalez de Vega et al. 2020). While matrix-dependent instrumental fractionation effects are not a major concern in this work, because all tektites and microtektites have glassy matrix, it is important to stress that MC-ICP-MS coupled with femtosecond laser ablation system would provide a major step forward in dealing with such unwanted effects [e.g. see Steinhofel et al., 2009, Chem. Geol. 268, 67–73. DOI:10.1016/j.chemgeo.2009.07.010].

In the “materials and methods” section we added extra sentences in line 453: “*High precision Fe isotope ratios determined in situ in the geosciences are often measured using SIMS.⁷² Although SIMS is at present characterized by superior spatial resolution and the obtained isotope ratio precision is nearly similar to that of LA-MC-ICP-MS, the latter is less prone to matrix effects or topographic features and has more flexibility for the instrumental mass discrimination correction. In this work,...*”

The reference 72 [Marin-Carbonne et al. 2011 Chem. Geol. 285, 50–61. DOI:10.1016/j.chemgeo.2011.02.019] was added to the bibliography list.

Overall, in this work SIMS might be a suitable alternative to nsLA-MC-ICP-MS, once the Fe isotopic analysis protocol is well established and validated.

Relatedly, I greatly appreciate your explaining the methods in greater detail and responding to my earlier concerns.

We appreciate you and referee #2 pointing out that such more elaborate description was needed, as it improved the quality of the manuscript, and also demonstrates the intrinsic value of LA-MC-ICP-MS to the scientific community.

The figure that you provided in the reviewer response may also be beneficial as a supplementary figure, if you think that would be appropriate and help other readers

We need to be extra careful to avoid double publication of the data in this manuscript and in an earlier report. The figure of the Fe isotope signatures of GRMs is based on the data already reported in Gonzalez de Vega et al. 2020, so re-publishing this might be considered to go against ethical codes, as the reference material data are not entirely new.

2) In the conclusion, there is discussion of GASP and HASP, which are specific to the Moon. I think these sentences can be moved up earlier, as there are other places where the Moon and lunar glasses are mentioned. In the conclusion, perhaps provide a summary sentence of comparisons, rather than introducing an entirely new concept here

We assume that with “in the conclusion”, the reviewer refers to the “Implications for Solar System processes” paragraph of the discussion section, in which we discuss the significance of the Fe isotope data in microtektites for our understanding of Solar System processes, and compare the observations of our work with complementary observations made for impact materials from the Moon and other planetary materials?

As this implications paragraph is not *per se* a conclusion of an entire manuscript, and is de facto a part of discussion section, I don't see a major problem with introducing a new concept here. The implications section compares the $\delta\text{Fe-FeO}$ patterns observed in this work with the available observations of impact materials in the Solar System to hypothesize what could be the effect of impact processing onto isotopic evolution of the target surfaces through the history of planetary bodies. While it is possible in principle to move the description of GASP and HASP into the introduction (The only other section in the manuscript where the Moon is mentioned), it would – in our opinion - result in an unnecessary disconnection of the logical narrative.

3) I find it interesting that chondrules have similar signatures. Though this may be beyond the scope of this work, is there any evidence that the conditions of chondrule formation may be similar to hypervelocity impacts?

The similar pattern in $\delta\text{Fe-FeO}$ signatures is probably because both materials experienced a sequence of evaporation and condensation. However, chondrules formed at very different conditions in the Early Solar Nebula, compared to the formation of microtektites. First of all, the temperature, pressure and timeframes of an expanding impact plume, where microtektites are formed, are completely different from those in gradually cooling protosolar nebulae. Additionally, the Earth is characterized by a highly oxidative and significant atmosphere (*i.e.* oxygen fugacity and pressure). In the original version of the paper, some inferences to nebular processes were made, but based on the suggestion of reviewer #1 these have been removed.

4) Depending on publication limits, if need be, Figures 5-7 could be moved to the supplement.

As pointed out by the editor, 10 items can be included in the publication, so there is no need to move the figures into the supplement, but we appreciate the reviewer for pointing this out.

6) For background, there are many references missing. To recognize that there are additional references, please use e.g. in these citations, for example in lines 49-51. There is no need to add references, especially given limits on space, but it is important to not cite references as if they are the definitive source for a statement.

All the Best!

We corrected the sentence in lines 49-51 according to the comment. However, it is increasingly difficult to add additional text to the manuscript due to word limits, as it requires shortening the text in other places.

Line 47-58: briefly describe heavy versus light isotopic signatures here. Could move lines 104-108 here

This is correct, the concept of shifting the isotope signatures by evaporation is only discussed later in the text. So we changed “*heavy isotopic signatures*” to “*altered isotopic signatures*” in line 47.

Line 55-58: switch sentences, so that it reads: “Such impacts...Tektites...”

The order of the sentences “such impacts...” and “tektites are...” was switched according to this comment.

Line 71: Please clarify, i.e. I think the South China Sea is part of the Australasian strewn field, but I am not sure?

The sentence in line 71 was corrected: “*The recovery of shock metamorphosed rock and unmelted mineral fragments in Australasian microtektite layers in the South China Sea...*” By definition, a strewn field implies the area over which impact materials, including the microtektite layers, are deposited, so we hope that the current phrasing infers that we are discussing the Australasian strewn field?

Line 140-141: Are petrographic observations from this study? If so, what are the signs of alteration? If not, then please reference the appropriate study.

To address this comment, we added a reference to the supplementary materials at the end of this sentence, which contains a more detailed description of potential signs of alteration.

Line 160: Please use “e.g.” to describe moderately volatile elements because there are more than are mentioned in this line.

The correction was made according to this comment.

Line 196: change “hint towards” to “indicate”

The wording was changed according to this suggestion.

Line 314-315: reference needed for lunar glasses

A reference to Warren (2008) on GASP and HASP was added here.

Line 282: This heading is a little unclear and vague, “Overprinting of the Fe isotope fractionation mechanisms,” The overprinting is by condensation, and should be specified as such. Additionally, the vapor phase that condenses is part of the Fe isotope fractionation mechanism. Perhaps “Overprinting of heavy Fe isotope signatures by light vapor condensate”?

The heading was changed according to the request.

Line 368: Is it possible to quantify these statements? “move fast” / “very quickly”

The sentence was re-written according to the comment: “*However, due to the supersonic velocity up to 8 km s^{-1} , microtektites are expected to lose such vaporized shell before it re-condenses.*”

Line 389-391: This scenario could be compared to differences in Cu versus Fe condensation, see Day et al., (2019)

This reference was added at the end of the sentence.